# How far can we go with ImageNet for Text-to-Image generation?

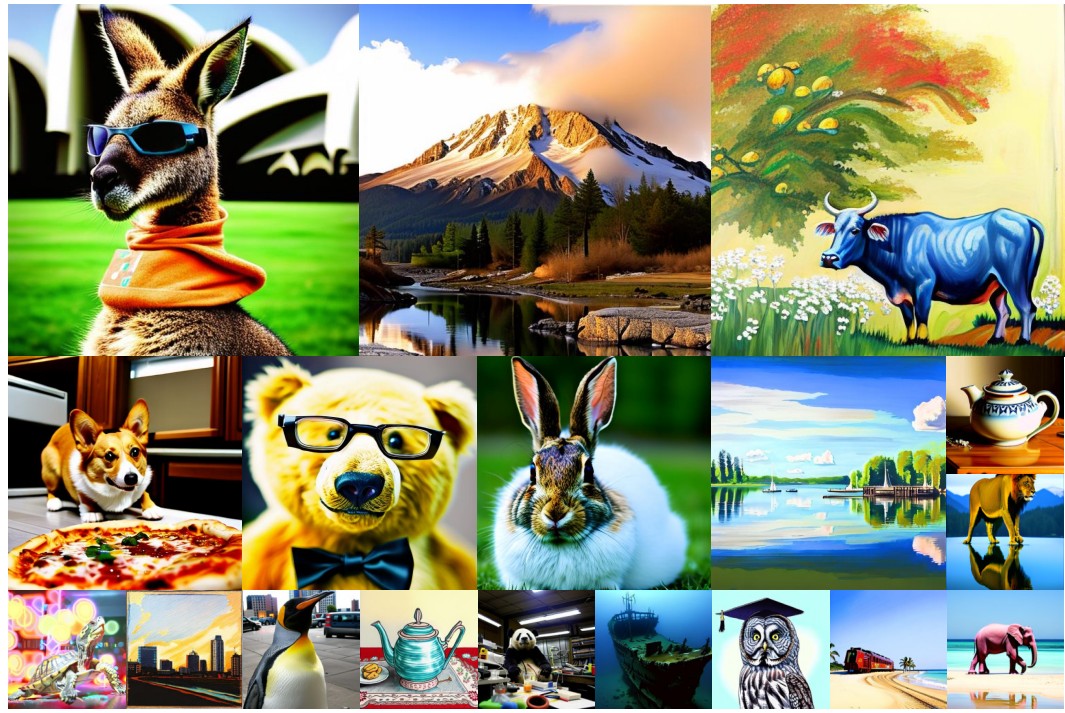

Figure 1: **Images** generated by our 400M parameters text-to-image model trained solely on ImageNet. Text prompts are taken from `PartiPrompts` [53].

## Abstract

Recent text-to-image (T2I) generation models have achieved remarkable results by training on billion-scale datasets, following a 'bigger is better' paradigm that prioritizes data quantity over availability (closed vs open source) and reproducibility (data decay vs established collections). We challenge this established paradigm by demonstrating that one can match or outperform models trained on massive web-scraped collections, using only ImageNet enhanced with well-designed text and image augmentations. With this much simpler setup, we achieve a $+1\%$ overall score over SD-XL on GenEval and $+0.5\%$ on DPGBench while using just *1/10th the parameters and 1/1000th the training images*. This opens the way for more reproducible research as ImageNet is a widely available dataset and our standardized training setup does not require massive compute resources.

# 1 Introduction

The prevailing wisdom in text-to-image (T2I) generation holds that larger training datasets inevitably lead to better performance. This "bigger is better" paradigm has driven the field to billion-scale image-text paired datasets like LAION-5B [43], DataComp-12.8B [10] or ALIGN-6.6B [36]. While this massive scale is often justified as necessary to capture the full text-image distribution, in this work, we challenge this assumption and argue that data quantity overlooks fundamental questions of data efficiency and quality in model training.

Our critique of the data-scaling paradigm comes from a critical observation: current training sets are either closed-source or rapidly decaying which makes results impossible to fully reproduce, let alone compare fairly. As such, the community of T2I generation is in dire need of a standardized training setup to foster open and reproducible research. Luckily, Computer Vision already has such dataset in ImageNet [41] that has been the gold standard in many tasks for many years. It is widely available and its strength and limitations are well known. Furthermore, it is heavily used in class-conditional image generation [35, 20], which makes its evaluation metrics more familiar. This begs the question of *how far can we go with ImageNet for text-to-image generation?*

Our findings are that we can indeed get a surprisingly competitive model by training solely on ImageNet. As shown on Figure 1, we can achieve excellent visual quality. Additionally we also achieve very competitive scores on common benchmarks such as GenEval [11] and DPGBench [18], matching or even surpassing popular models that are trained on much more data and at a far greater cost, such as SDXL [37] and Pixart-$\alpha$ [3] (see Figure 2). However, this does not come without any hurdles. In this paper we analyze the challenges of training using ImageNet only and propose successful strategies to overcome them. Our strategies allow us to train models of smaller size (about 300M parameters) on a reasonable compute budget (about 500 H100 hours) making it accessible to more research teams, while not compromising on the capabilities.

Our contributions are thus the following:

- We analyze the shortcomings of training T2I diffusion models on ImageNet and propose mitigation strategies.
- Then, we propose a standardized training setup using only images from ImageNet, providing accessible and reproducible research for T2I generation.
- We provide several models in the 300M-400M parameters range generating high quality images and outperforming competing models that are 10 times the size and trained on 1000 times more data.

To commit to open and reproducible science, all our training data are hosted at `https://huggingface.co/datasets/anonymous_for_review` and all our code and models are hosted at `https://github.com/anonymous_for_review`.

In the next section, we outline the challenges in using ImageNet for T2I generation, and then evaluate mitigation strategies for each of them. We then combine them in a complete training recipe that we use to train several models of varying resolution. We compare them against the state of the art, with excellent results. Finally, we discuss the related work before we conclude.

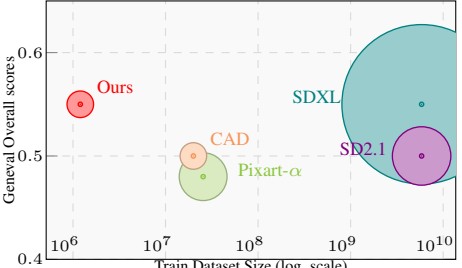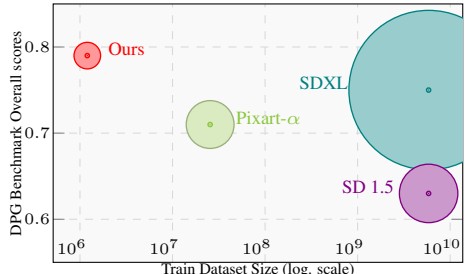

Figure 2: **Quantitative results** on GenEval (left) and DPGBench (right). The size of the bubble represents the number of parameters. In both cases, we outperform models of $10\times$ the parameters and trained on $1000\times$ the number of images.

| Model | TA | FID Inc. IN Val↓ | FID Inc. COCO↓ | Jina CLIP IN Val↑ | Jina CLIP COCO↑ | Prcesion↑ | Recall↑ | Density↑ | Coverage↑ | GenEval Overall↑ |
|---|---|---|---|---|---|---|---|---|---|---|
| DiT-I | ✗ | 20.14 | 71.00 | 31.21 | 22.42 | 0.67 | 0.29 | 0.71 | 0.39 | 0.11 |
|  | ✓ | 6.29 | 45.71 | 38.45 | 38.39 | 0.77 | 0.76 | 0.82 | 0.72 | 0.55 |
| CAD-I | ✗ | 84.77 | 46.35 | 20.55 | 14.06 | 0.75 | 0.05 | 1.40 | 0.10 | 0.17 |
|  | ✓ | 6.16 | 49.89 | 38.01 | 37.85 | 0.80 | 0.72 | 0.89 | 0.76 | 0.55 |

Table 1: **Image quality and compostionality** of AIO models and TA models. FID reported is FID `Inception v3`. Precision, Recall, Density and Coverage are computed using `DINOv2` features on ImageNet Val. Values on COCO test set are reported in Table 8.

## 2 Adopting ImageNet for Text-to-Image generation

We focus on training text-to-image models using ImageNet, a small, open-source and widely accepted data collection. We first discuss the evaluation criterions and then gradually pinpoint the major limitations in setting up a T2I diffusion model using ImageNet. To overcome these limitations, we show that well-crafted augmentations can bring forth a compositionally accurate T2I model with data constraints. For our analysis, we leverage two architectures: (1) **DiT-I** (our adaptation of DiT [35] to handle text) and (2) **CAD-I** [6]. The suffix "I" is added to indicate the model is being trained only on ImageNet.

**Evaluation: *Image-quality*.** We specifically assess the generation quality of both in-distribution (w.r.t Imagenet-50k validation set) and zero-shot (MSCOCO-30k captions validation set [30]). Specifically, we adopt: **(1) FID** [16] using both standard `Inception-v3` and `Dino-v2` backbones, (2) **Precision** [27], (3) **Recall** [27], (4) **Density** [34], and (5) **Coverage** [34]. These are all calculated with `Dino-v2` features.

**Evaluation: *Compositionality*.** To understand the text-image alignment capabilities and image composition prowess, we adopt (1) **CLIPScore** [15] and (2) **Jina-CLIP Score** [26] on both MSCOCO-30k and ImageNet validation set, (3) **GenEval** [11] and (4) **DPGBench** [18].

**Evaluation: *Aesthetic-quality*** to assess the human aesthetics of generated images. In line with prior works [9], to assess the aesthetical understanding of the generated images we also adopt: (1) **PickScore** [25], (2) **Aesthetics Score** [43], and (3) **VILA Score** [22] using `PartiPrompts` [53].

Based on these evaluation strategies that assess both image quality and compositional accuracy, we can systematically identify the key limitations of training T2I diffusion models on ImageNet. Our analysis reveals several ImageNet-specific challenges that must be addressed to achieve high-performance generation with limited data.

### 2.1 Text challenges

**Challenge: *Absence of captions*.** Class-conditional models trained with ImageNet have shown exceptional generation capabilities [35, 33] However, extending this use to T2I generation is difficult since ImageNet, being a classification dataset, lacks any sort of caption corresponding to its images. To adopt ImageNet for T2I generation, similar to prior works [38], one could build captions by a very simple strategy of `'An image of <class name>'` (denoted AIO). However, this results in very poor generation capabilities as shown in Table 1. This can be mainly attributed to the two major shortcomings of AIO captions for ImageNet: First, AIO captions lack vocabulary. They contain only roughly a thousand words corresponding to the concepts of the classes and thus lack attributes, spatial relations, etc. This constraint on the diversity in the text-condition space leads to a clear bottleneck in text understanding. Second, there is often more content in the image than just the class. For example, a caption "an image of *golden retriever*" mentions the class name but leaves out details and concepts that could be in the background. This lack of details leads to spurious correlation where the model can learn to associate unrelated visual pattern (e.g., grass texture) to the class name (e.g., *golden retriever*) because the text for this concept is never mentioned in the text space. Finally, despite the presence of humans in the images, ImageNet does not contain a '*person*' class, resulting in humans not being represented in the AIO text space. This issue extends to many categories (*road*, *water*, etc), as ImageNet is an object-centric dataset.

| Model | IA | Overall↑ | One obj.↑ | Two obj.↑ | Count.↑ | Col.↑ | Pos.↑ | Col. attr.↑ |
|-------|-----|----------|-----------|-----------|---------|-------|-------|-------------|
| | × | 0.55 | 0.95 | 0.61 | 0.36 | 0.80 | 0.28 | 0.33 |
| DiT-I | Crop | 0.54 | 0.96 | 0.56 | 0.38 | 0.79 | 0.22 | 0.33 |
| | CutMix | 0.57 | 0.96 | 0.68 | 0.36 | 0.74 | 0.28 | 0.39 |
| CAD-I | × | 0.55 | 0.97 | 0.60 | 0.42 | 0.74 | 0.26 | 0.35 |
| | CutMix | 0.57 | 0.94 | 0.68 | 0.40 | 0.70 | 0.35 | 0.36 |

Table 2: **GenEval scores** of TA and TA + IA models. All models are trained with long captions. A Prompt Extender was used before generating images. Models are evaluated at $256^2$ resolution.

**Solution:** ***Long informative captions.*** To overcome this challenge, we employ a synthetic captioner [32] (TA for *Text Augmentation*) to generate comprehensive captions that capture: (i) *Scene composition* and *spatial relationships*; (ii) *Background elements* and *environmental context*; (iii) *Secondary objects* and *participants*; (iv) *Visual attributes* (color, size, texture); and (v) *Actions* and *interactions* between elements.

We compare the gains attributed to long captions both quantitatively (Table 1) as well as qualitatively (Figure 4: row 1-2). For ImageNet-Val set, we observe that models trained with long captions significantly improves performance, resulting in lower FIDs of **6.29** for DiT-I and **6.16** for CAD-I in contrast to **20.14** for DiT-I and **85** for CAD-I on AIO captions. As a point of reference, we remind the reader that models of this size (below 0.5B parameters) typically have an FID of 9 using the class-conditional setup [35]. The aesthetic metrics (PickScore, Aesthetic Score, and VILA Score), offer additional insights, highlighting the superiority of TA models, which consistently outperform their AIO counterparts. For COCO test set - which is a zero-shot task for our training, this trend is all the more dramatic. The TA models are the only ones able to correctly follow the prompt as attested by the much improved Jina CLIP score (DiT-I from **22.42** to **38.45**; CAD-I from **14.06** to **37.85**), while keeping similar image quality.

Regarding text-image alignment and compositionality, models trained with longer captions benefit from the added information, evidenced by the improvement in GenEval overall score from (DiT-I from **0.11** to **0.55**; CAD-I from **0.17** to **0.55**) (see Table 1).

## 2.2 Image challenges

Using long, informative captions (TA) significantly enhances both generation quality and compositional alignment. But training text-to-image diffusion models on ImageNet only still faces two critical limitations: early overfitting and poor compositional generalization.

**Limitation:** ***Early overfitting.*** Models trained on ImageNet with long captions (TA) demonstrate promising initial performance. However, due to the relatively small scale of ImageNet (only 1.2 million images) they begin overfitting at approximately 200k training steps (see Figure 3).

**Limitation:** ***Restricted Complex Compositionality Abilities.*** ImageNet's object-centric nature presents a challenge for learning complex compositions. Even with enhanced textual descriptions via TA captioning, models still struggle with spatial relationships, attribute binding, and multi-object compositions. This limitation manifests in lower GenEval scores for compositional prompts involving multiple objects, color attribution, and positional relationships, as shown in Figure 4

**Solution:** ***Image Augmentation (IA).*** To reduce overfitting and improve compositional reasoning, we investigate the use of image augmentations during training. We experiment with two augmentation strategies. Details about implementation and training pipeline are given in Appendices E and F.

- **CutMix** [55]: For each image in the dataset, we randomly select an image from a different class and overlay a smaller version of it onto the original image. A caption is generated using the CutMix image as input. This technique introduces additional variability in the training data.

- **Crop**: During training, we randomly mask a portion of the image tokens such that the model is exposed only to a local square crop of the original image. We add some crop coordinates tokens to the captions of the image. This augmentation encourages the model to decouple

| Model | IA | FID Inc. IN Val↓ | FID Inc. COCO↓ | Jina CLIP IN Val↑ | Jina CLIP COCO↑ | Prcesion↑ | Recall↑ | Density↑ | Coverage↑ |
|---|---|---|---|---|---|---|---|---|---|
| DiT-I | × | 6.29 | 45.71 | 38.45 | 38.39 | 0.77 | 0.76 | 0.82 | 0.72 |
| | Crop | 6.20 | 44.04 | 38.45 | 38.39 | 0.77 | 0.75 | 0.83 | 0.74 |
| | CutMix | 7.30 | 49.12 | 38.77 | 36.80 | 0.79 | 0.74 | 0.88 | 0.75 |
| CAD-I | × | 6.16 | 49.89 | 38.01 | 37.85 | 0.80 | 0.72 | 0.89 | 0.76 |
| | CutMix | 6.62 | 49.31 | 38.17 | 37.71 | 0.80 | 0.70 | 0.90 | 0.76 |

Table 3: **Image quality** of TA models and TA + IA models. All models are trained with long captions. FID reported is FID `Inception v3`. Precision, Recall, Density and Coverage are computed using `DINOv2` features on ImageNet Val. Values on COCO test set are reported in Table 8.

| Model | IA | PickScore↑ | Aes.Score↑ | VILA↑ |
|---|---|---|---|---|
| DiT-I | × | 20.74 | 5.29 | 0.31 |
| | Crop | 20.63 | 5.17 | 0.30 |
| | CutMix | 20.81 | 5.34 | 0.31 |
| CAD-I | × | 20.03 | 5.17 | 0.28 |
| | CutMix | 20.03 | 5.16 | 0.28 |

Table 4: **Aesthetic metrics** of TA models and TA + IA models. All models are trained with long captions. Text prompts are taken from `PartiPrompts` [53]

object features from their background context, and to learn correspondences between partial visual elements and specific text tokens.

In Figure 3, we plot the evolution of FID and GenEval scores over training steps for the CAD-I architecture. Training with TA alone leads to early overfitting: we observe a sharp rise in FID after 200k steps. In contrast, models trained with TA+CutMix or TA+Crop maintain significantly lower FID curves for longer, with a delayed onset of overfitting.

Table 2 assesses the impact of image augmentation on GenEval metrics. Image augmentation (both CutMix and Crop) leads to a notable improvement in the GenEval overall score, with an increase of 2 points. Notably, the Two Objects sub-task sees a +7 point increase for both architectures, CAD-I sees a +9 point gain in Position, and DiT-I gains +6 points in Color Attribution.

These improvements in compositional metrics are achieved while maintaining or improving image quality, as measured by FID scores and Aesthetic metrics in Table 3. The qualitative examples in Figure 4 (rows 2-3) further demonstrate the enhanced compositional capabilities, with models trained using augmentation techniques producing more accurate representations of complex prompts. This improvement is particularly evident in the pirate ship scene: while the TA model generates a ship awkwardly positioned with a bowl of soup, the TA + IA model creates a more natural composition with the pirate ship appropriately sailing in the bowl. Similarly, the hedgehog and hourglass example shows more refined details and aesthetically pleasing composition with TA + IA, whereas the TA model struggles to render a recognizable hedgehog.

## 2.3 Scaling to higher resolution

All experiments discussed thus far were conducted at a resolution of $256^2$. We then explore whether high-resolution generation is feasible under the same data constraints. We investigate if this setup—utilizing only ImageNet with text and image augmentations—can scale to a $512^2$ resolution without compromising performance or requiring additional supervision.

We take a DiT-I checkpoint at $256^2$ resolution, trained with TA + IA for 250,000 steps, and continue training it at $512^2$ resolution for an additional 100,000 steps on the same data. Both the pretraining and the fine-tuning use the Crop image augmentation strategy for simplicity. This fine-tuning procedure requires no changes to the text encoder or transformer backbone, aside from adjusting the image tokenization to handle the larger input size.

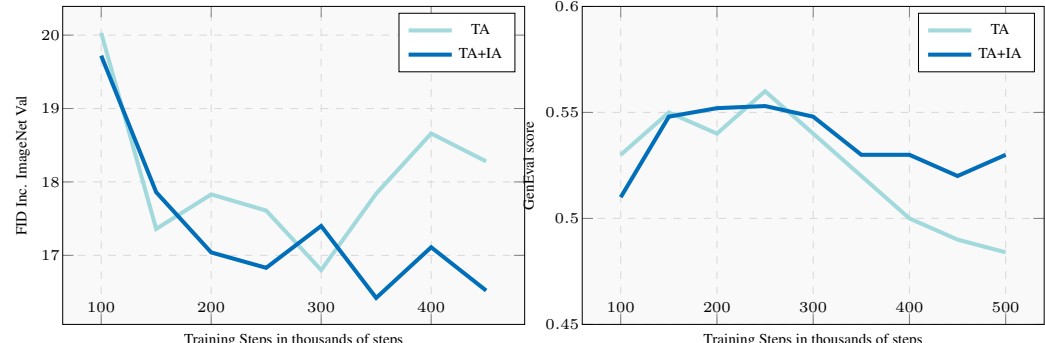

Figure 3: **Training dynamics showing FID and GenEval scores vs training steps**. TA + IA (navy blue) maintains better scores throughout training compared to TA only (light blue), demonstrating improved resistance to overfitting. Lower FID scores indicate better image quality. Better GenEval scores indicate better compositionality abilities.

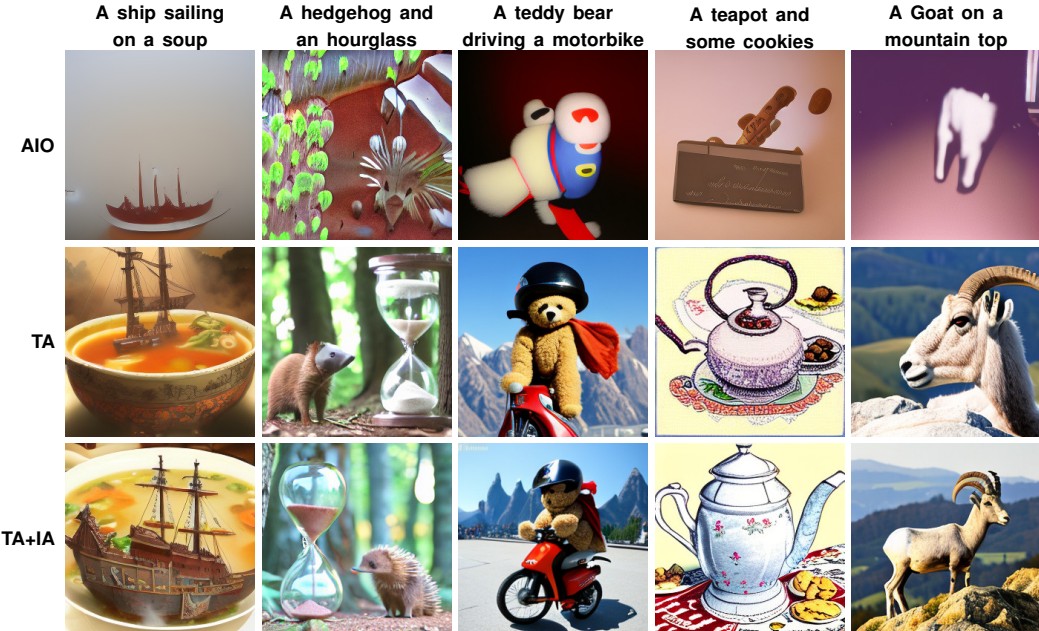

Figure 4: **Qualitative comparison across models**. From top to bottom: 'An image of {class-name}' (AIO), Text-Augmentation (TA), and Text-Augmentation with Image-Augmentation (TA + IA) models. The examples show generated images of (from left to right): *a pirate ship sailing on a steaming soup*, *a hedgehog and an hourglass*, *a teddy bear riding a motorbike*, *a teapot sitting on a decorative tablecloth* and *a goat on a mountains*. While text augmentation improves the model's understanding, image augmentation leads to better text comprehension and higher image quality overall.

Figure 1 presents qualitative samples from the $512^2$ model across a wide range of prompts taken from `PartiPrompts` [53]. Compared to its $256^2$ counterparts, the model produces images with noticeably sharper details, more saturated colors. The higher-resolution model's GenEval scores are reported in Table 5. It exhibits slight but consistent gains on the GenEval benchmark over its $256^2$ initialization (Table 2 line 2). In particular, we observe improvements on the Counting (+5 points compared to the $256^2$ Crop model; see Table 2) and Two Objects (+8 points) sub-tasks.

## 3 Comparison with SOTA

In this section, we report the results of our best models and compare them against the state of the art.

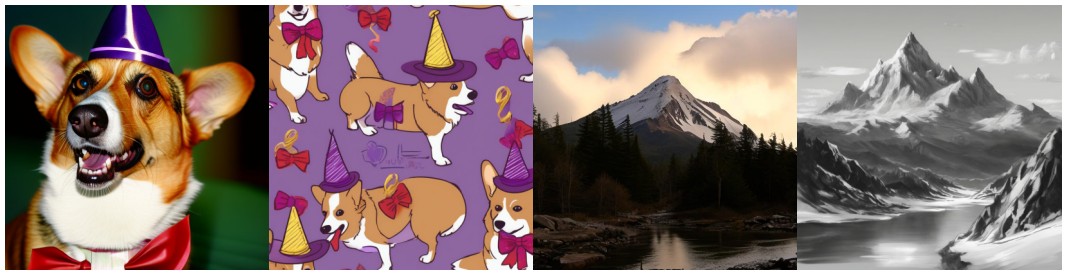

A corgi wearing a red bowtie
and a purple party hat.

A mountain

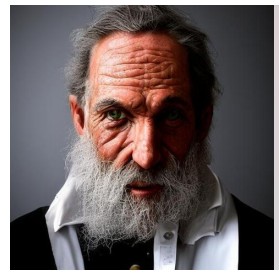 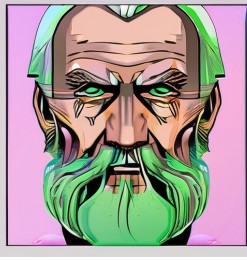 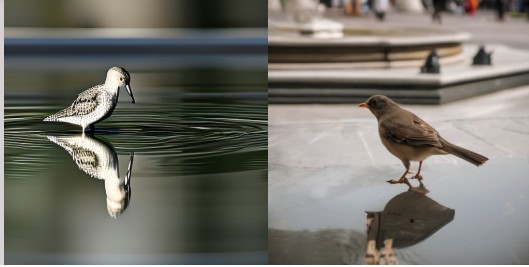

An old man with a long
grey beard and green eyes

A bird and its reflection in a fountain.

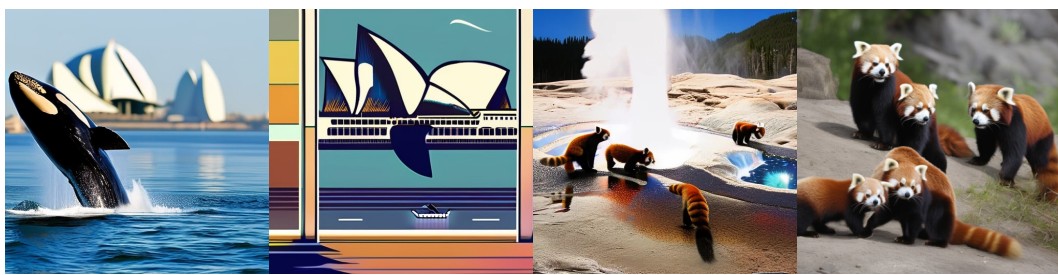

A whale breaching in front
of the Sydney Opera House

A family of red pandas
passing by the geyser

Figure 5: **Qualitative comparison**. Pairs of images generated with our DiT-I model (left) and SDXL (right). Each pair shows the same prompt rendered by both models. We match the visual quality of the popular SDXL without the need to train on a massive web-scraped dataset.

**Qualitative results.** On Figure 5, we show a comparison of images generated by our 512 DiT-I model and SDXL. We are able to achieve similar visual quality as SDXL, with a tendency to generate more photorealistic images. In contrast, SDXL is more likely to generate synthetic images (drawings, painting, digital art). Given that ImageNet is overwhelmingly composed of photograph, this difference of bias is likely to come from the dataset. In addition, we are slightly better at handling prompt following (e.g., on the first image, only a single corgi, with the correct hat color ; lack of reflection on the bird in the fountain ; lack of geyser on the last image).

**Quantitative results: Comparison to the state of the art on GenEval and DPG benchmarks.** We test the composition ability of both our best DiT-I and CAD-I models, trained with TA + IA, on the GenEval and DPGBench benchmarks and compare our performances to the ones of popular state-of-the-art models.

**GenEval.** Table 5 reports the results on GenEval benchmark. We observe that our high-resolution 512 model (**0.56**) performs better on average than SD1.5 (**0.43**), Pixsart-$\alpha$ (**0.48**), SD2.1 (**0.50**) and

| Resolution | Model | Nb of params | Training set size | Overall↑ | One obj.↑ | Two obj.↑ | Count.↑ | Col.↑ | Pos.↑ | Col. attr.↑ |
|---|---|---|---|---|---|---|---|---|---|---|
| Native resolution | SD v1.5 | 0.9B | 5B+ | 0.43 | 0.97 | 0.38 | 0.35 | 0.76 | 0.04 | 0.06 |
|  | PixArt-$\alpha$ | 0.6B | 0.025B | 0.48 | **0.98** | 0.50 | **0.44** | 0.80 | 0.08 | 0.07 |
|  | SD v2.1 | 0.9B | 5B+ | 0.50 | **0.98** | 0.51 | **0.44** | **0.85** | 0.07 | 0.17 |
|  | SDXL | 3.5B | 5B+ | 0.55 | **0.98** | **0.74** | 0.39 | **0.85** | 0.15 | 0.23 |
| $512^2$ | DiT-I (Ours) | 0.4B | 0.001B | **0.56** | 0.94 | 0.64 | 0.43 | 0.77 | **0.25** | **0.35** |
| $256^2$ | SD v1.5 | 0.9B | 5B+ | 0.13 | 0.48 | 0.04 | 0.01 | 0.23 | 0.00 | 0.00 |
|  | PixArt-$\alpha$ | 0.6B | 0.025B | 0.48 | **0.96** | 0.51 | **0.48** | 0.78 | 0.07 | 0.08 |
|  | CAD | 0.4B | 0.020B | 0.50 | 0.95 | 0.56 | 0.40 | 0.76 | 0.11 | 0.22 |
| $256^2$ | DiT-I (Ours) | 0.4B | 0.001B | **0.58** | 0.95 | 0.67 | 0.43 | **0.80** | 0.30 | 0.35 |
|  | CAD-I (Ours) | 0.3B | 0.001B | 0.57 | 0.94 | **0.68** | 0.40 | 0.70 | **0.35** | **0.36** |

Table 5: **Results on GenEval**. The top section represents results reported with a native resolution of $512^2$ or above. In the bottom section, models are evaluated at $256^2$ resolution. **Bold** indicates best, underline second best.

| Model | Params | Training set size | Global↑ | Entity↑ | Attribute↑ | Relation↑ | Other↑ | Overall↑ |
|---|---|---|---|---|---|---|---|---|
| SDv1.5 | 0.9B | 5B+ | 74.63 | 74.23 | 75.39 | 73.49 | 67.81 | 63.18 |
| Pixart-$\alpha$ | 0.6B | 25M | 74.97 | 79.32 | 78.60 | 82.57 | 76.96 | 71.11 |
| CAD | 0.4B | 20M | 84.50 | 85.25 | 84.66 | 91.53 | 74.8 | 77.55 |
| SDXL | 3.5B | 5B+ | 83.27 | 82.43 | 80.91 | 86.76 | 80.41 | 74.65 |
| SD3-Medium | 2B | 1B+ | **87.90** | **91.01** | **88.83** | 80.70 | **88.68** | **84.08** |
| Janus | 1.3B | 1B+ | 82.33 | 87.38 | 87.70 | 85.46 | 86.41 | 79.68 |
| DiT-I $256^2$ CutMix (Ours) | 0.4B | 1.2M | 82.07 | 85.61 | 84.59 | 91.41 | 74.8 | 77.5 |
| DiT-I $256^2$ Crop (Ours) | 0.4B | 1.2M | 81.46 | 84.71 | 86.00 | 92.71 | 74.8 | 76.34 |
| CAD-I $256^2$ CutMix (Ours) | 0.3B | 1.2M | 80.85 | 87.48 | 85.32 | **93.54** | 78.00 | 79.94 |
| DiT-I $512^2$ (Ours) | 0.4B | 1.2M | 79.94 | 83.21 | 83.42 | 90.14 | 72.0 | 75.14 |

Table 6: **Results on DPG-Bench**. We compare our models to the results reported in [52]. **Bold** indicates best, underline second best.

SDXL (**0.55**) in their native resolution. The striking improvements of our model are in the position attribute and color attribution where our model achieves more than **+10** w.r.t SDXL. In *lower* $256^2$ resolution the scores are even more distinctive as our models (CAD-I: **0.57**; DiT-I: **0.58**) outperform all the other popular models (SD1.5 (**0.13**), PixArt-$\alpha$ (**0.48**), and CAD (**0.50**)). In both $256^2$ and $512^2$ resolution our models reaches the performance of SDXL in its native resolution while *having 10x fewer parameters and being trained on only 0.1% of the data.*

**DPGBench.** Table 6 reports the results on DPGBench, a recent benchmark similar to Geneval but with a more complex prompt set. We observe similar trends as for GenEval: compared to the current leaderboard, we achieve an overall accuracy of **75.1%** with DiT-I, which improves over SDXL by **0.5%**. We reach an overall score of **79.94%** for CAD-I, outperforming SDXL by **+5%** and PixArt-$\alpha$ by **+8%**. Impressively, our models reach accuracies comparable to that of Janus [52], a 1.3B parameters VLM with generation capabilities. Notably, both our models are particularly good at relations, achieving state-of-the-art of 93.5% for CAD-I and 92.2% for DiT-I. Even at higher $512^2$ resolution our model achieves better overall score (**75.14%**) than SDXL (**74.65%**), even with only $100k$ steps of high-resolution fine-tuning.

## 4 Related Work

**Diffusion Models.** [46, 17, 45] have demonstrated remarkable success across various domains [19, 4, 7]. While image generation remains their most prominent application [5, 46, 21], text-to-image (T2I) synthesis [40, 42, 39] has emerged as a particularly impactful use case. These models operate by learning to reverse a gradual Gaussian noise corruption process. At extreme noise levels, the model effectively samples from a standard normal distribution to produce realistic images. The core optimization objective is:

$$\min_\theta \mathbb{E}_{(x_0,c)\sim p_{\text{data}}, \epsilon\sim\mathcal{N}(0,1)} \left[ \|\epsilon - \epsilon_\theta(x_t, c, t)\|^2 \right] \tag{1}$$

where $x_t = \sqrt{\gamma(t)}x_0 + \sqrt{1-\gamma(t)}\epsilon$ denotes the noised image at timestep $t$, $x_0$ the original image, $c$ the corresponding condition (such as text), $\epsilon$ is standard normal noise, $\epsilon_\theta$ the learned noise predictor, and $\gamma(t)$ the variance schedule.

**Computational Efficiency.** Traditional diffusion models require substantial computational resources, with leading implementations consuming hundreds of thousands of GPU hours [40]. Recent advances have significantly improved training efficiency. [51, 54] identified limitations in the diffusion loss's representation learning capabilities, demonstrating that supplementary representation losses accelerate convergence. [3] achieved dramatic compute reduction by repurposing class-conditional models for text-to-image generation. [6] introduced architectural improvements and coherence-aware mechanisms, matching Stable Diffusion's performance [40] with 100x fewer GPU hours.

**Data Efficiency.** Early T2I models relied on billion-scale web-scraped datasets [40], creating accessibility barriers due to storage requirements and reproducibility challenges from copyright restrictions. [3] pioneered dataset reduction using 20M high-quality images from recaptioned SAM data [24], though portions remain proprietary. Subsequent work explored CC12M [2, 13, 6] and YFCC100M's public subset [47, 12], revealing overfitting below 10M samples. Our approach diverges by leveraging ImageNet [41] – a reproducible, well-established benchmark with standardized metrics [16]. We transform this classification dataset into T2I training data through synthetic captions and image augmentations.

**Synthetic captions.** Synthetic image captioning has benefited several tasks. For instance, visual question answering [44] and visual representation learning [48] achieve state-of-the-art performances by enhancing the captioning output of Vision-Language Models (VLMs) [28, 44]. Similarly, training with synthetic captions for text-to-image generation is becoming the defacto protocol for large diffusion models, such as DALL-E [1], Pixart-$\alpha$ [3] and Stable Diffusion-3 [8]. More recently, some approaches [31, 29] extend this approach by training text-to-image (T2I) models on multi-level captions. Inspired by these, our method deploys the popular LLaVA captioner [32] to augment existing textual captions and use them to train text-to-image generation models.

# 5 Conclusion

In this work, we challenged the prevailing wisdom that billion-scale datasets are necessary for high-quality text-to-image generation. These large-scale datasets are usually either closed sourced or rapidly decaying, which threatens openness and reproducibility in text-to-image generation research. Instead, we show that it is possible to train smaller models to high quality using ImageNet only. We analyze the shortcomings of ImageNet for text-to-image generation and propose data-augmentation strategies to overcome these limitations.

We thus propose a standardize text-to-image training setup that leads to models capable of generating high quality images while being excellent at prompt following. This is attested by results on common benchmarks ($56\%$ or $+1\%$ on GenEval and $75.1\%$ or $+0.5\%$ on DPGBench) on which we are able to outperform popular models such as SDXL.

The implications of our work extend beyond just computational efficiency, open science and reproducibility. By showing that smaller datasets can achieve state-of-the-art results, we open new possibilities for specialized domain adaptation where large-scale data collection is impractical. Our work also suggests a path toward more controllable and ethical development of text-to-image models, as smaller datasets enable more thorough content verification and bias mitigation.

Looking forward, we believe our results will encourage the community to reconsider the "bigger is better" paradigm. Future work could explore additional augmentation strategies, investigate the theoretical foundations of data efficiency, and develop even more compact architectures optimized for smaller datasets. Ultimately, we hope this work starts a shift toward more sustainable and responsible development of text-to-image generation models.

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

# A   Additional Qualitative Results at $256^2$ resolution

## A.1   Based on ImageNet validation set captions

DiT-I                         CAD-I

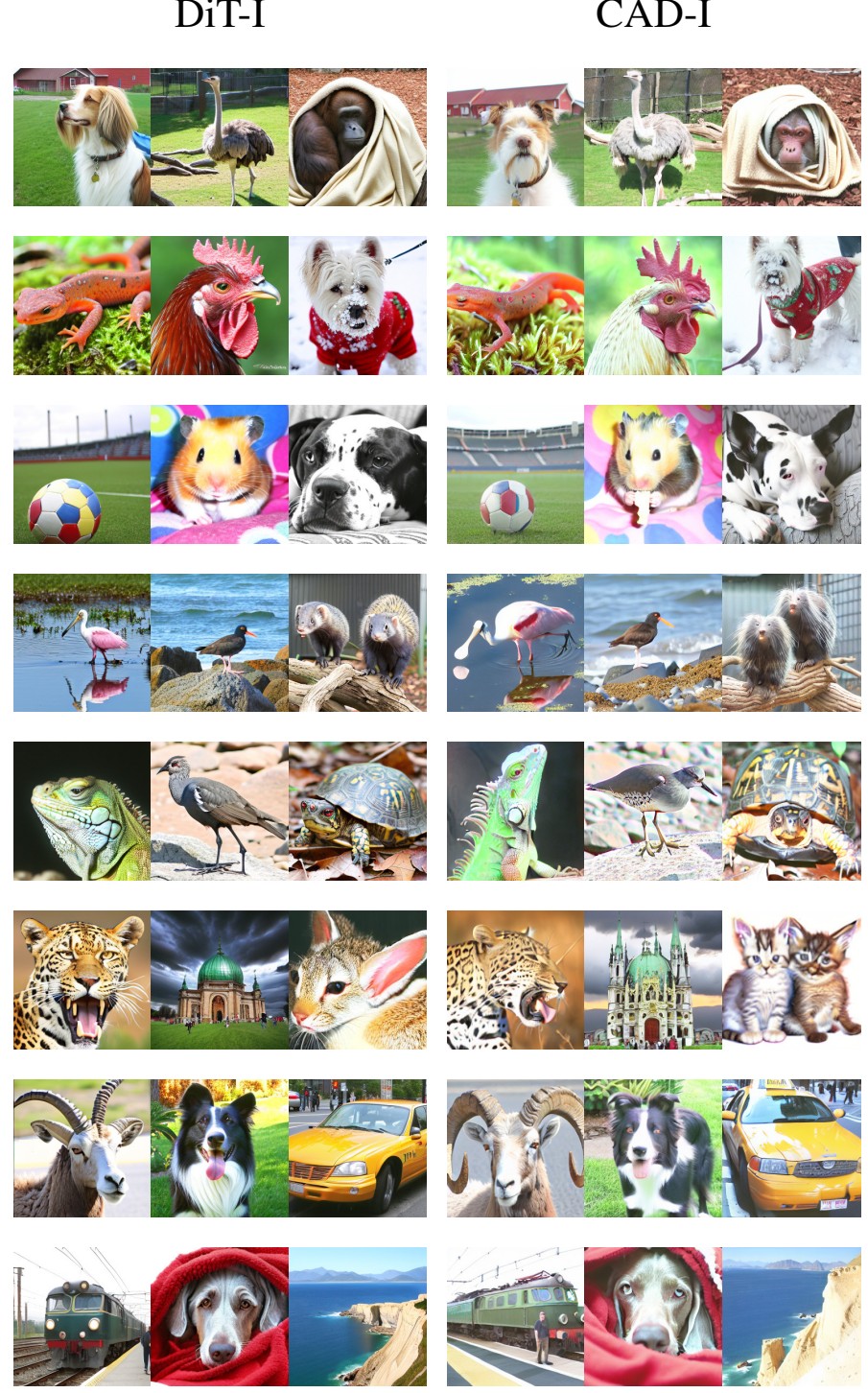

 **A.2   Based on DPG bench captions**

DiT-I                                        CAD-I

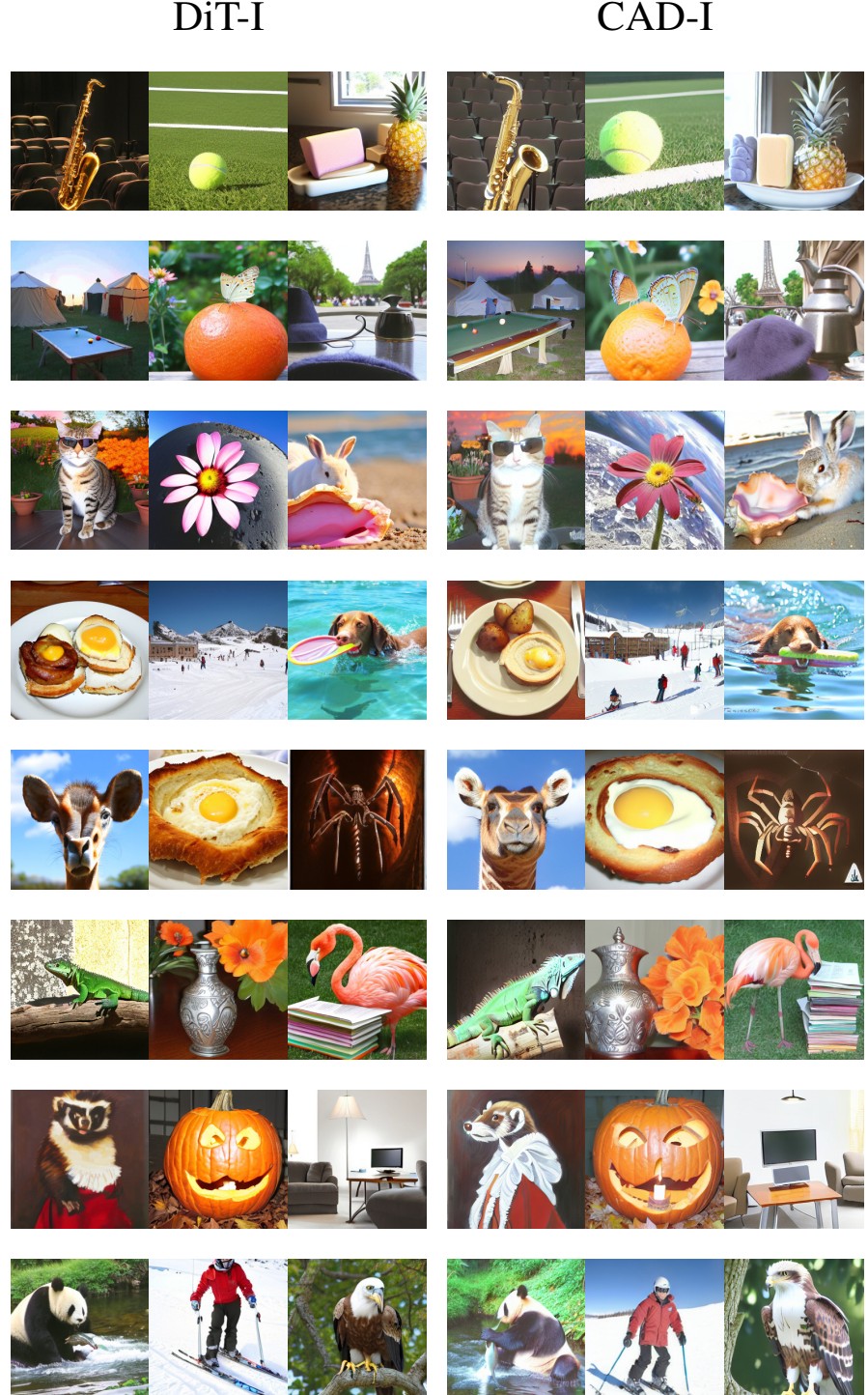

 # B  Additional quantitative results

| Model | TA | IA | ImageNet Val 50k | | | | | | | |
|-------|----|----|--------------|-------------|------|------|------|------|------|--------|
| | | | FID Inc.↓ | FID DINOv2↓ | P↑ | R↑ | D↑ | C↑ | CS↑ | Jina-CS↑ |
| DiT-I | ✗ | ✗ | 20.14 | 356.13 | 0.67 | 0.29 | 0.71 | 0.39 | 23.98 | 31.21 |
| | ✓ | ✗ | 6.29 | 90.31 | 0.77 | 0.76 | 0.82 | 0.72 | 9.08 | 38.63 |
| | ✓ | Crop | 6.20 | 89.03 | 0.77 | 0.75 | 0.83 | 0.74 | 9.25 | 38.45 |
| | ✓ | CutMix | 7.30 | 83.71 | 0.79 | 0.74 | 0.88 | 0.75 | 9.36 | 38.77 |
| CAD-I | ✗ | ✗ | 84.77 | 904.50 | 0.75 | 0.05 | **1.40** | 0.10 | 8.94 | 20.55 |
| | ✓ | ✗ | **6.16** | **91.53** | **0.80** | 0.72 | 0.89 | **0.76** | 8.69 | 38.01 |
| | ✓ | CutMix | 6.62 | 91.72 | **0.80** | 0.70 | 0.90 | **0.76** | 8.53 | 38.17 |

Table 7: **Ablation study** on ImageNet dataset. We use prompts in accordance with the training setup (*i.e.*, "An image of <class>" for short captions and a caption generated from the validation images for long captions). Precision, Recall, Density and Coverage are computed using `DINOv2` features. **Bold** indicates best, underline second best

| Model | TA | IA | COCO 30k | | | | | | | |
|-------|----|----|--------------|-------------|------|------|------|------|------|--------|
| | | | FID Inc.↓ | FID DINOv2↓ | P↑ | R↑ | D↑ | C↑ | CS↑ | Jina-CS↑ |
| DiT-I | ✗ | ✗ | 71.00 | 1107.22 | 0.46 | 0.07 | 0.37 | 0.08 | 18.03 | 22.42 |
| | ✓ | ✗ | 45.71 | 631.91 | 0.64 | 0.44 | 0.52 | 0.29 | 25.52 | 36.63 |
| | ✓ | Crop | 44.02 | 627.78 | 0.63 | 0.41 | 0.51 | 0.29 | 25.46 | 38.39 |
| | ✓ | CutMix | 49.12 | 631.01 | 0.65 | 0.45 | 0.54 | 0.30 | 25.68 | 36.80 |
| CAD-I | ✗ | ✗ | 46.35 | 858.43 | 0.52 | 0.18 | 0.45 | 0.15 | 12.89 | 14.06 |
| | ✓ | ✗ | 46.93 | 655.37 | **0.66** | **0.42** | **0.61** | 0.28 | 26.37 | 35.72 |
| | ✓ | CutMix | 49.41 | **646.51** | **0.66** | 0.41 | 0.57 | **0.29** | **26.60** | **36.51** |

Table 8: **Ablation study** on COCO dataset. Precision, Recall, Density and Coverage are computed using `DINOv2` features. **Bold** indicates best, underline second best

# C   Implementation details

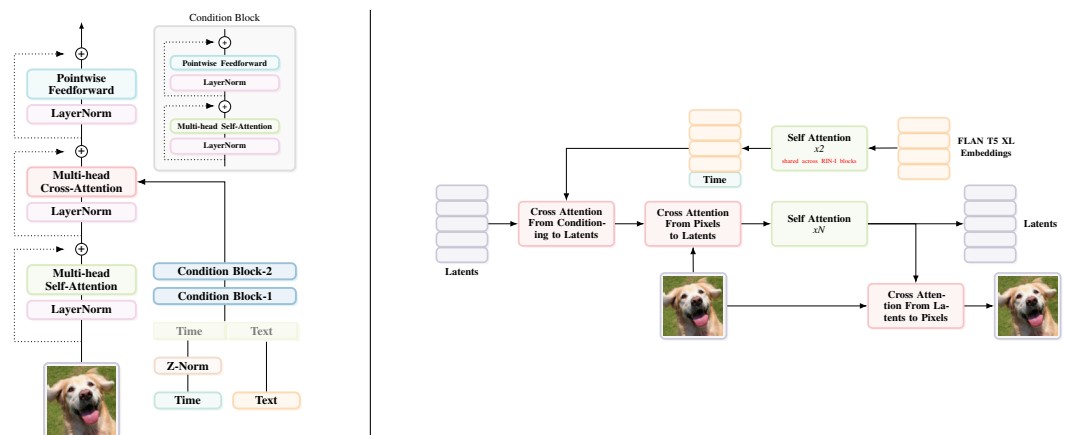

Figure 6: **Fundamental architecture blocks used in our experiments**. *Left*: DiT-I block and *Right*: CAD-I block.

In this work, we use both DiT [35] and RIN [20] architectures. To adapt DiT for text-conditional setting, we replace `AdaLN-Zero` conditioning with *cross-attention* to input the text condition into the model, as in [3]. Before feeding the text condition to the model, we refine it using *two* self-attention layers. Similar to [8], we add QK-Normalization [14] in each of the *self-attention* and *cross-attention* blocks to mitigate sudden growths of attention entropy and reduce training loss instability. We also add LayerScale [49] to each of the residual blocks of DiT for further stability. Figure 6 details our DiT-I architecture.

To adapt the RIN [20] for the text-conditional setting, we used the off-the-shelf architecture from [6], an adaptation of the RIN architecture detailed in the Appendix of [6]. Figure 6 details our CAD-I architecture.

We use the framework of latent diffusion [40]. For encoding the images into the latent space, we use the pre-trained variational autoencoder [23, 50] provided by the authors of Stable Diffusion [40]. The checkpoint used is available on HuggingFace: `https://huggingface.co/stabilityai/sd-vae-ft-ema`. For text conditions, we encode the captions using the T5 text encoder. The checkpoint is available on HuggingFace: `https://huggingface.co/google/flan-t5-xl`.

## D  Captioning details

**Captioning efficiently with LLaVA**  To caption images, we use the checkpoint `llama3-llava-next-8b-hf` (available on HuggingFace: `https://huggingface.co/llava-hf/llama3-llava-next-8b-hf`) with the prompt *"Describe this image"*. LLaVA encodes images using a dynamic resolution scheme. It processes both the entire image and four distinct patches as unique images and concatenates them. For 256x256 images, LLaVA uses around 2500 image tokens. To make the captioning process more efficient, we prune the image tokens, retaining only the tokens of the entire image and discarding patch-specific tokens. This optimization increased inference speed by a factor of 2.7, without compromising performances. Examples of long captions generated by LLaVA are given in Figure 7.

**Captioning CutMix images**  We caption CutMix images from $\text{CM}^{1/2}$ with similar settings used for captioning the original ImageNet images. However, to ensure that LLaVA does not describe both the base and the CutMix images independently, we use a different prompt: *"Describe this image. Consider all the objects in the picture. Describe them, describe their position and their relation. Do not consider the image as a composite of images. The image is a single scene image"*.
For settings $\text{CM}^{1/4}$, $\text{CM}^{1/9}$ and $\text{CM}^{1/16}$, LLaVA tends to either ignore the smaller CutMix image or describe the image as a composite of two images. To avoid this behaviour, we encode the image by using the entire image patch and add tokens from the patch to which the CutMix image belongs. We use the following prompt: *"Describe this image. Consider all the objects in the picture. Describe them, describe their position and their relation. Do not consider the image as a composite of images. The image is a single scene image"*. Examples of long captions generated by LLaVA for CutMix images are given in Figure 7.

## E  CutMix details

The CutMix framework systematically combines concepts while preserving object centrality. Our framework defines four precise augmentation patterns, each designed to maintain visual coherence while introducing novel concept combinations. These are briefly described below:

1. **CM$^{1/2}$** (Half-Mix):
   *Scale:* Both images maintain their original resolution.
   *Position:* Deterministic split along height or width.
   *Coverage:* Each concept occupies 50% of final image.
   *Preservation:* Both concepts maintain full resolution.

2. **CM$^{1/4}$** (Quarter-Mix):
   *Scale:* CutMix image resized to 50% side length.
   *Position:* Fixed placement at one of four corners.
   *Coverage:* 2nd concept occupies 25% of final image.
   *Preservation:* Base image center region remains intact.

3. **CM$^{1/9}$** (Ninth-Mix):
   *Scale:* CutMix image resized to 33.3% side length.
   *Position:* Fixed placement along image borders.
   *Coverage:* 2nd concept occupies 11.1% of final image.
   *Preservation:* Base image center, corners remain intact.

4. **CM$^{1/16}$** (Sixteenth-Mix):
   *Scale:* CutMix image resized to 25% side length.
   *Position:* Random placement not central 10% region.
   *Coverage:* 2nd concept occupies 6.25% of final image.
   *Preservation:* Base image center region remains intact.

Each augmentation strategy generates 1,281,167 samples, matching ImageNet's training set size. Figure 7 shows examples of the different structured augmentations.

We also define **CM$^{all}$**, which uniformly samples from all four patterns. The CM$^{all}$ variant combines equal proportions (25%) from each pattern to maintain the same total sample count. Post-augmentation, we apply LLaVA captioning to all generated images, ensuring semantic alignment between visual and textual representations. This produces detailed descriptions that accurately reflect the augmented content while maintaining natural language fluency.

### E.1  Training with CutMix images

Because the CutMix image augmentations have strong artefacts corresponding to the boundaries of the mixing, we have to prevent the model from learning those salient features and reproducing them. To that end, we propose to train on image augmentation only at timesteps $t$ where the noisy image $x_t$ is sufficiently noisy that the artifacts no longer matter. In practice, this corresponds to sampling either from the original image training set $\mathcal{A}$ or from the augmented image training set $\mathcal{A}_{\text{IA}}$ conditionally to $t$, compared to an additional hyperparameter $\tau$ deciding whether $t$ is sufficiently large for image augmentation. This extra condition leads to replacing the original diffusion loss in Equation 1 with

$$\min_{\theta} \mathbb{E}_{\substack{t \sim \mathcal{U}(0,T), \\ \rho \sim \mathcal{B}_{(\tau,p)}(t), \\ (x,c) \sim \mathcal{A}(\rho), \\ \epsilon \sim \mathcal{N}(0,1)}} \left[ \|\epsilon - \epsilon_{\theta}(x_t, c, t)\|^2 \right]. \tag{2}$$

In this novel loss, the timestep $t \sim \mathcal{U}(0, T)$ is still sampled uniformly. We introduce a new random variable $\rho$ that is sampled conditionally to $t$, where $\mathcal{B}_{\tau,p}(t)$ denotes a specific distribution that corresponds to:

$$\mathcal{B}_{\tau,p}(t) = \begin{cases} 0, & \text{if } t \leq \tau, \\ \mathcal{B}_p, & \text{else.} \end{cases} \tag{3}$$

Here, $\mathcal{B}_p$ a Bernoulli distribution of parameter $p$. The text-image pair $(x_0, c)$ is then sampled conditionally to $\rho$, where $\mathcal{A}(\rho)$ is a distribution that uniformly samples from the original or the augmented datasets depending on $\rho$:

$$\mathcal{A}(\rho) = \begin{cases} \mathcal{A}, & \text{if } \rho = 0, \\ \mathcal{A}_{\text{IA}}, & \text{else.} \end{cases} \tag{4}$$

| Setting | Long Caption |
|---|---|
| **LLaVA w/o CM** 
 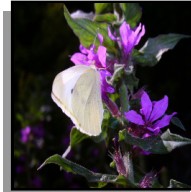 | In the heart of the verdant background, a delicate white butterfly with a hint of yellow around its edges is captured in mid-flight. Its wings, a blend of two shades of cream and white, add a soft contrast to the vibrant colors of the flower it's hovering over. The flower, a stunning shade of purple, is adorned with a yellow center that speaks of the sun's rays it might have been basking in. Surrounding it is a lush green foliage, adding depth to this tranquil scene. The butterfly, positioned slightly to the left of the flower's center, seems to be in the process of alighting or perhaps taking off, adding a dynamic element to this otherwise serene tableau. |
| **CM$^{1/2}$** 
 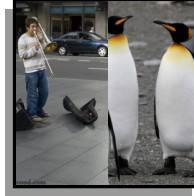 | On the left side, there is a person playing the trumpet on a street. The individual is standing on the ground, wearing a long-sleeved shirt and jeans. On the ground next to the player, there is a trash can. The street is made of asphalt, with buildings in the background and a clear sky overhead. On the right side of the image, there are two penguins standing on a gravel surface, possibly a road or a rocky beach. The penguins are frozen in place, displaying a contrast to the dynamic scene on the left side. The penguins' yellow beaks, black bodies, and white feathers are clearly visible. The image captures an everyday moment of music played on the street, juxtaposed with the surreal and unexpected presence of the penguins. The image does not provide information about the time of day, specific location, or any actions being performed by the penguins or the person. The image is static, presenting two moments captured in time rather than a sequence of events. |
| **CM$^{1/4}$** 
 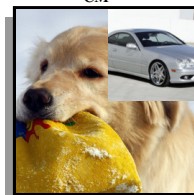 | The image shows a large, muscular dog in the foreground and a silver sports car in the background. The dog, appearing to be a Golden Retriever, is on the left side of the frame, occupying a significant portion of the space. It has a yellow hue with darker markings, and its ears are floppy. The dog is heavily coated with snowflakes, suggesting that the photo was taken in a snowy environment. The sports car, positioned on the right, is a two-door coupe with sleek curves and a notable design, featuring the Mercedes-Benz logo on its front grille. The car has a silver finish, and the photo captures it from a perspective that shows the front and side profile. The car is parked on an asphalt surface, possibly a parking lot or a driveway. The dog is facing the camera with direct gaze, while the car is positioned slightly towards the side, away from the viewer's perspective. |
| **CM$^{1/9}$** 
 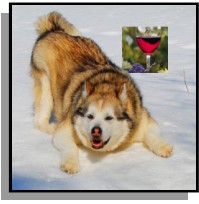 | The image depicts a picturesque outdoor scene featuring an ornate building, which appears to be a palace or manor house, with classical architectural elements including symmetrical windows, a central cupola, and multiple chimneys. In front of the building is a well-maintained garden with pathways and neatly trimmed hedges or borders. Above the garden, there is a clear blue sky with a few scattered clouds. In the sky, there is a single hot air balloon with a bright orange and yellow pattern. The balloon is floating at a considerable height above the garden and the building, suggesting it might be part of a leisure activity or a special event. The image is a photograph with natural lighting, indicative of a sunny day. |
| **CM$^{1/16}$** | The image is a photograph featuring a husky dog resting in the snow. The dog has a light coat with darker markings around its face and ears, and it is lying on its side with its head up, looking directly at the camera. Its eyes are open and its mouth is slightly open, showing teeth and a pink tongue, which suggests the dog might be panting or in a relaxed state. Next to the dog's side, there is a wine glass with red wine and a few purple flowers, which could be lilacs, positioned on the left side of the glass stem. The wine glass and flowers are set against a blurred background that gives the impression of greenery. |

Figure 7: **Long captions generated by our synthetic LLaVA captioner**. The captions generated are highly diverse and add in much more intricate details of *compositionality*, *colors* as well as *concepts* which are not present in the original ImageNet dataset. The captions generated for our augmented images are also highly coherent and explain the scene in a much more realistic way.

498 The noise $\epsilon$ is sampled from the Normal distribution, as in the usual diffusion equation. Similarly, the
499 noisy image $x_t$ is obtained by $x_t = \sqrt{\gamma(t)} x_0 + \sqrt{1 - \gamma(t)} \epsilon$.
500 This novel loss function is more involved than the regular diffusion training; yet, in practice, it is
501 very easy to implement and can be done entirely during the mini-batch construction as described in
502 Algorithm 1.
503 Figure 8 illustrates our complete CutMix pipeline.

**Algorithm 1** Batch with CutMix image augmentation

---
1: **Input:** dataset $\mathcal{A}, \mathcal{A}_{\text{IA}}$, augmentation time $\tau$, augmentation probability $p$, batch size $m$
2: $B \leftarrow \{\}$
3: **for** $i = 1$ **to** $m$ **do**
4:     $t \sim \mathcal{U}(0, T)$
5:     $(x_0, c) \sim \mathcal{A}$
6:     **if** $t > \tau$ **then**
7:         $\rho \sim \mathcal{B}_p$
8:         **if** $\rho$ **then**
9:             $(x_0, c) \sim \mathcal{A}_{\text{IA}}$
10:        **end if**
11:    **end if**
12:    $\epsilon \sim \mathcal{N}(0, 1)$
13:    $x_t = \sqrt{\gamma(t)}x_0 + \sqrt{1 - \gamma(t)}\epsilon$
14:    $B \leftarrow B \cup \{(x_t, c, t)\}$
15: **end for**
16: **Return:** $B$

---

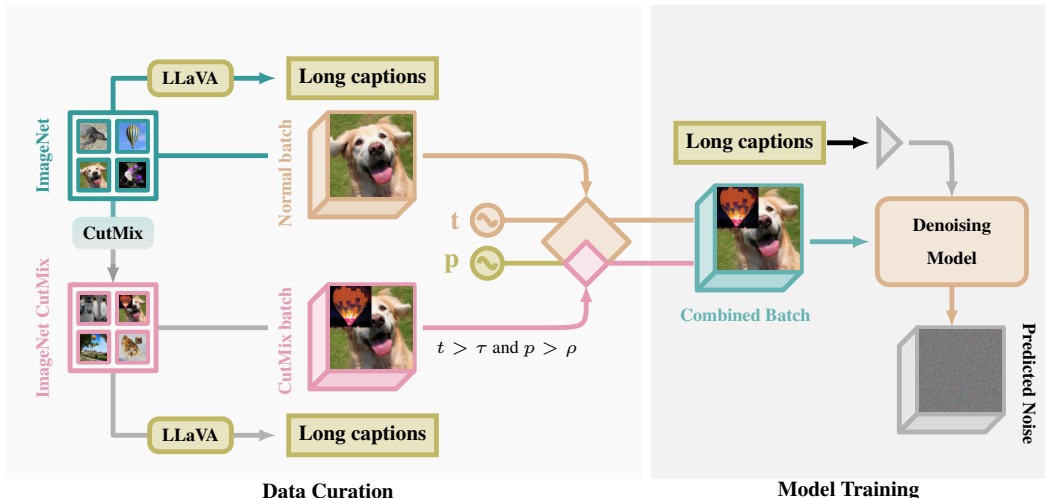

Figure 8: **Pipeline of our Cutmix Data Curation and Training process.** Starting from ImageNet, we a) use LLaVa VLM to caption the images into long detailed caption (top branch left) and b) use several CutMix strategies to create new images combining several ImageNet concepts and caption them using LLaVa into long and detailed captions (bottom branch left). During training, we sample batches of normal and CutMix images and we select from each batch depending on the timestep $t$ at which the CutMix strategy is valid and a probability $p$ of sampling CutMix images.

## E.2 CutMix Augmentation Ablations

### E.2.1 Ablation on CutMix settings

First, we analyse the performances of the pixel augmentations for $\{\mathbf{CM}^{1/2}$ , $\mathbf{CM}^{1/4}$ , $\mathbf{CM}^{1/9}$ , $\mathbf{CM}^{1/16}$ , $\mathbf{CM}^{all}\}$ settings. We fix the probability of using a pixel-augmented image in the batch when $t > \tau$ to $p = 0.5$ and we measure both image quality and composition ability. Results are reported in Table 9.

For image quality, all settings seem to perform similarly, with $\mathbf{CM}^{1/2}$ being the best at 6.13 FID and $\mathbf{CM}^{all}$ being the worst at 6.81 FID. This indicates that all settings are able to avoid producing uncanny images that would disturb the training too much.

For composition ability, $\mathbf{CM}^{1/16}$ can improve over the baseline on extended prompts, whereas $\mathbf{CM}^{all}$ can improve over the baseline on original prompts. Overall, only $\mathbf{CM}^{all}$ manages to keep closer performances between the original prompts and the extended ones. Since $\mathbf{CM}^{all}$ is a mixture of

all other settings, it also has the most diverse training set and is thus harder to overfit. As such, we consider $\mathbf{CM}^{all}$ for the best models.

| Model | CutMix Settings | FID↓ | GenEval↑ | |
|---|---|---|---|---|
| | | | ◇ | ⋆ |
| CAD-I | $\mathrm{CM}^{1/2}$ | 6.13 | 0.46 | 0.55 |
| | $\mathrm{CM}^{1/4}$ | 6.41 | 0.49 | 0.53 |
| | $\mathrm{CM}^{1/9}$ | 6.63 | 0.51 | 0.51 |
| | $\mathrm{CM}^{1/16}$ | 6.42 | 0.47 | 0.56 |
| | $\mathrm{CM}^{all}$ | 6.81 | 0.53 | 0.55 |

Table 9: **Ablation study** on CutMix settings. The probability of sampling CutMix images used here is $\rho = 0.5$. Models are trained for 250k steps. FID is computed on the ImageNet val set with long prompts, using the `Inception-v3` backbone. ◇ means original GenEval prompts. ⋆ means extended GenEval prompts.

| Model | $\rho$ | FID↓ | GenEval↑ | |
|---|---|---|---|---|
| | | | ◇ | ⋆ |
| CAD-I | 0 | 6.16 | 0.51 | 0.55 |
| | 0.25 | 5.99 | 0.55 | 0.58 |
| | 0.5 | 6.41 | 0.49 | 0.53 |
| | 0.75 | 6.71 | 0.45 | 0.53 |
| | 1 | 6.07 | 0.48 | 0.49 |

Table 10: **Ablation study** on probability $\rho$ of sampling a CutMix image during training. The CutMix setting is $\mathrm{CM}^{1/4}$. Models are trained for 250k steps. FID is computed on ImageNet val set with long prompts, using the `Inception-v3` backbone. ◇ means original GenEval prompts. ⋆ means extended GenEval prompts.

| Model | $\tau$ | FID↓ | GenEval↑ | |
|---|---|---|---|---|
| | | | ◇ | ⋆ |
| CAD-I | 300 | 6.99 | 0.51 | 0.53 |
| | 400 | 6.62 | 0.55 | 0.57 |
| | 500 | 6.16 | 0.48 | 0.55 |
| | 600 | 5.90 | 0.50 | 0.55 |

Table 11: **Ablation study** on timestep threshold $\tau$. The CutMix setting is $\mathbf{CM}^{all}$ . Models are trained for 250k steps. FID is computed on ImageNet val set with long prompts, using the `Inception-v3` backbone. ◇ means original GenEval prompts. ⋆ means extended GenEval prompts.

### E.2.2 Ablation on CutMix probability

Next, we analyse the influence of the probability $p$ of using a pixel augmented image in the batch, when the condition on $t$ is met. Results for $p \in \{0.25, 0.5, 0.75, 1.0\}$ are shown in Table 10, using $\mathbf{CM}^{1/4}$ pixel augmentations.

As we can see in terms of image quality, the FID is slightly degraded by having too frequent pixel augmentation ($p > 0.5$). This can be explained by the fact that pixel-augmented images are only seen when $t > \tau$. As such, a high value for $p$ creates a distribution gap between the images seen for $t > \tau$ and the images seen for $t \leq \tau$.

Composition ability shows a similar behaviour with the GenEval overall score decreasing when $p$ increases for both the original and the extended prompts. As such, we consider $p \leq 0.5$ for the best models.

### E.2.3 Ablation on threshold $\tau$

Finally, we analyse the influence of the threshold $\tau$, which enables CutMix images to be sampled in training batches. Table 11 shows the FID Inception on ImageNet Val and the GenEval scores of models trained with different $\tau$ values.

We find that $\tau = 400$ results in the highest GenEval score of $0.55$ on original prompts and $0.57$ on extended prompts, while $\tau = 600$ yields the lowest FID on ImageNet Val. As such, we use $\tau = 400$ for the best models.

## F  Cropping details

Our cropping training methodology (see Figure 9) removes spurious concept correlations due to its masking scheme. We maintain the original captions and force the model to independently identify relevant textual elements. This creates a more challenging learning task for the model that enhances text-image alignment. During training, we only consider tokens corresponding to small portion of the image and mask out the rest from both the loss function and cross-attention layers. Given we do this online, this is highly efficient and also allows an infinite training set based on ImageNet to train on. For making the model understand the full dynamics of the training data, in our training scheme with crops, we only feed cropped images to the model with a probability, $p = 0.5$. In rest of the cases, we use entire image. To keep the training scheme as simple as possible, for cropped versions, we only use crop resolution of >50% of the normal resolution.

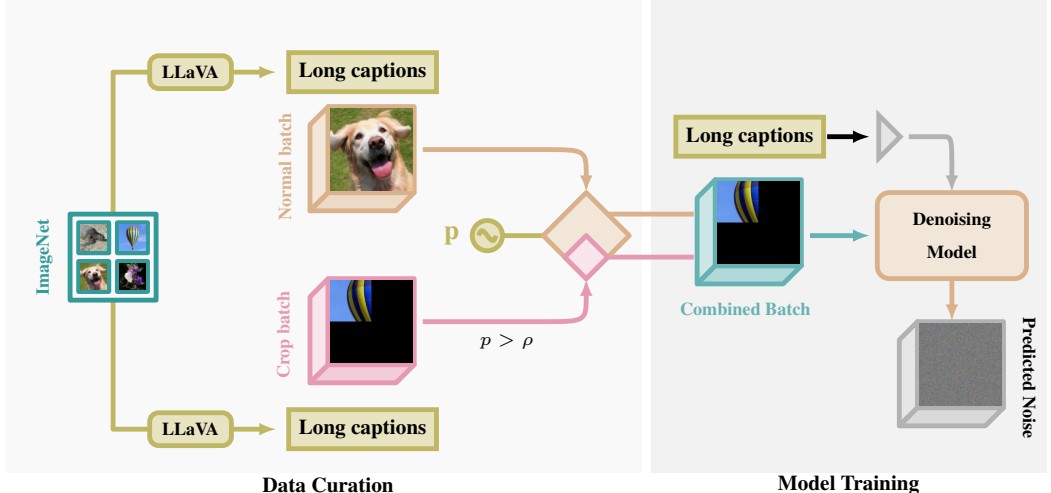

Figure 9: **Pipeline of our Cropping Data Curation and Training process.** Starting from ImageNet, we a) use LLaVa VLM to caption the images into long detailed caption and b) use cropping strategies to create new images from ImageNet by cropping. We keep the same captions as if we were using the original image. During training, we do cropping online with a probability $p$ of sampling cropped images. The crop images can have any resolution >50% of the original resolution.

# G Qualitative results prompts

Here we show the prompts used to make the Figure 4. Note that for AIO we use the short version of the prompt as it is closer to its train distribution:

1. **A pirate ship sailing on a streaming soup**

    The image showcases a colossal, exquisitely crafted pirate ship, its presence commanding and larger-than-life, as it sails triumphantly across a boundless sea of steaming soup. The ship's hull, made of dark, polished wood, is adorned with intricate carvings of dragons and waves, while its three towering masts support vast, billowing sails that glow faintly in the warm, golden light radiating from the broth. The soup is a vibrant, aromatic masterpiece, with swirls of rich broth, floating islands of noodles, and vibrant vegetables like carrots, bok choy, and mushrooms creating a textured, immersive landscape. The ship's deck is alive with detail—ropes coiled neatly, barrels stacked high, and a crow's nest peeking above the sails, all slightly damp from the soup's rising steam. The bowl, an enormous, ornate vessel, is crafted from gleaming porcelain, its surface painted with delicate, hand-drawn scenes of mountains and rivers, adding a layer of cultural richness to the surreal composition. The scene is both absurd and breathtaking, blending the grandeur of a seafaring adventure with the comforting, whimsical charm of a bowl of soup, creating an image that is unforgettable and endlessly imaginative.

2. **A hedgehog and an hourglass**

    The image features a small, brown hedgehog with its characteristic spiky coat, standing near an hourglass in the middle of a dense forest. The hourglass is made of clear glass, and fine grains of sand are visible as they fall from the top chamber to the bottom. The forest surrounding the hedgehog and the hourglass is lush and green, with tall trees and thick undergrowth. Sunlight filters through the leaves, creating dappled patterns on the forest floor. The scene evokes a sense of tranquillity and the passage of time. The hedgehog appears to be observing the falling sand, perhaps contemplating the fleeting nature of time.

3. **A teddy bear riding a motorbike**

    A plush teddy bear, adorned with a shiny black motorcycle helmet and a flowing red cape, is perched confidently on a miniature red motorcycle. The toy bike and its adventurous rider are positioned against the bustling backdrop of Rio de Janeiro, with the iconic Dois Irmãos mountain peaks rising majestically in the distance. The scene captures the playful contrast between the soft texture of the teddy bear and the sleek metal of the motorcycle, all under the bright Brazilian sun.

4. **A teapot and some cookies**

    A detailed illustration of a teapot sitting on a decorative tablecloth, with delicate floral patterns and intricate stitching. The teapot itself has a sturdy handle and a gleaming silver spout, emitting a gentle steam as if freshly poured. The surrounding table features a few scattered tea leaves, and a plate with a few cookies, adding a touch of warmth and coziness to the scene. The illustration style is whimsical, with bold lines and vibrant colors, creating a sense of playfulness and inviting the viewer to take a sip from the teapot.

5. **A goat on a mountain top**

    A detailed photograph of a majestic goat standing atop a rocky outcropping, its white coat speckled with patches of brown and its curved horns reaching towards the sky. The goat's eyes are alert, and its ears are perked up, as if listening to some distant sound. In the background, a serene landscape unfolds, with rolling hills and a distant mountain range, all bathed in soft, warm sunlight that casts gentle shadows across the goat's fur.

