# OpenReview forum: "How far can we go with ImageNet for Text-to-Image generation?"
_NeurIPS.cc/2025/Conference — Submitted to NeurIPS 2025_

### Official Review · Reviewer_fyFz · 2025-06-14

**Clarity:** 4
**Significance:** 3
**Originality:** 3
**Rating:** 5
**Confidence:** 4

**Summary:**

This paper introduces a training paradigm with data augmentation for T2I models using only the ImageNet dataset. The approach can be extended to generate high-resolution images, demonstrating that effective performance does not require massive datasets or models with billions of parameters. The proposed model achieves competitive results on benchmarks like GenEval and DPG-Bench, outperforming popular models such as SDXL and PixArt-$\alpha$ despite having a smaller model size and training dataset.

**Questions:**

* Could the proposed approach be applied directly to other datasets to improve data efficiency more broadly? Given the known limitations of ImageNet, starting with a richer or more diverse dataset might lead to even better results.
* Although CutMix is applied only during noisy timesteps, using images that are significantly different may still introduce artifacts. Are there any limitations in the current approach that might lead to such issues, and how are they addressed?
* How does the model handle concepts that are rare or absent in ImageNet, such as abstract scenes, complex interactions, or human-centric prompts?
* I’m also curious about how CutMix contributes to learning spatial relationships in complex scenes. Could the approach be extended to involve more than two objects during augmentation (three or four) to further enhance compositional capabilities?



(Since the reviewer understands that training a T2I from scratch is time-consuming and might not be on time within the rebuttal timeline, it is acceptable for the authors to provide text-only explanations.)

**Ethical Concerns:**

["NO or VERY MINOR ethics concerns only"]

**Final Justification:**

After considering the authors' rebuttal, I find that the paper has sufficiently addressed previous concerns and has reached a level appropriate for acceptance. Specifically, the following points have been clearly addressed:

1. The disadvantages of CutMix have been clearly articulated, and the authors provided a convincing strategy to mitigate these drawbacks through second-stage fine-tuning at higher resolutions.
2. Initially, the experiments were primarily conducted on ImageNet. However, following the rebuttal, the authors expanded their evaluation to include results on ImageNet+COCO and LAION-pop, demonstrating robust performance across these datasets.
3. Previously noted limitations regarding the simplicity or singularity of objects in the examples have been clarified by the authors, who have now pointed to the provided composite examples.

(The authors should ensure that these rebuttal points, including those raised by other reviewers, are incorporated into the final manuscript to ensure completeness.)

Although the authors present limited examples of adapting their method to small-scale datasets and do not fully explore using subsets of large-scale datasets to generate diverse prompts, these are minor concerns compared to the overall contribution. The paper offers valuable insights into efficiently obtaining competitive models with reduced effort. Regarding the concerns about novelty raised by other reviewers, I believe that leveraging existing methodologies does not diminish the significance of the contribution, especially given the new insights provided. Hence, my final rating is to accept this paper.

**Limitations:**

While the authors acknowledge the limitations of using the ImageNet dataset, they do not discuss the limitations of their proposed approach. Here are some from the reviewer's perspective:
1. The CropMix augmentation may disrupt visual aesthetics from a human perspective. A more refined augmentation method could be developed to enhance compositional ability while preserving visual appeal.
2. Since the method relies on data augmentation from external models (e.g., text augmentation using LLaVa), establishing a validation protocol would help ensure the quality and accuracy of the augmented data.

**Quality:**

4

**Strengths And Weaknesses:**

### **Strengths**

* The paper is well-organized and clearly written. The motivation behind the work is effectively communicated, with detailed explanations that guide the reader.
* The topics addressed are timely and relevant, especially in the context of the growing scale of models and datasets.
* The experimental setup is clearly presented, and the results are thoroughly explained, making the conclusions easy to follow.

### **Weaknesses**

* Although the authors aim to challenge the notion that "bigger is better," the results still somewhat support this idea. For instance, in Table 6, models like SD3 and Janus outperform the proposed method in various aspects. While the proposed model achieves the second-highest overall score, Janus surpasses it in several specific categories. Additionally, techniques like data augmentation—used in this work—effectively expand the data space, indirectly reinforcing the idea that "more is better." A tighter focus on data efficiency alone might strengthen the central message.
* The proposed method is only evaluated on ImageNet, without testing on other widely used datasets such as LAION or DataComp. Applying the same training pipeline to a small, randomly sampled subset (e.g., 0.001B samples) from these datasets could help validate the generalizability and robustness of the approach.
* The timing and conditions for applying CutMix are critical, yet these details are included only in Appendix E-1. Moving this information to the main paper would improve clarity and help alleviate concerns about potential artifacts introduced by CutMix.
* Most of the provided images depict either single objects or multiple objects from the same class. It is recommended to include results showcasing more complex multi-object scenarios to better demonstrate the model’s text-to-image (T2I) generation capabilities.

---

### **Overall Evaluation**

Overall, I lean toward accepting this paper. The contributions are timely and relevant, and the results are promising. However, the paper would benefit from a clearer articulation of its position with respect to the "bigger is better" narrative, as well as more comprehensive experimental details. Addressing these points would further strengthen the paper's impact and clarity.

---

> ### Author Rebuttal · Authors · 2025-07-31
>
> ## TL;DR: Concise Summary of Our Revisions
>
> Our work demonstrates that competitive text-to-image models can be trained on a small, accessible dataset like ImageNet, challenging the need for massive commercial datasets.
>
> **Key Contributions & New Experiments:**
>
> * **Data-Efficiency:** We introduce a new metric, $\text{score}_{\text{eff}} = \frac{\text{score}}{\log(\text{dataset size})}$, to show our models achieve superior performance relative to their training data size, outperforming massive baselines.
> * **Generalizability:** We successfully applied our approach to other small datasets, finding that ImageNet surprisingly yields better results than LAION-POP, confirming its value as a standardized and accessible training source.
> * **Multi-object Capabilities:** We will add more examples of complex, multi-object scenarios to the final paper to better showcase the model's capabilities.
> * **Novel Concepts:** New experiments show our model can generate high-quality images for concepts not in ImageNet (e.g., "iPad") by leveraging the text encoder and reinterpreting existing visual patterns.
> * **CutMix Limitations:** We acknowledge potential caption artifacts from our CutMix strategy and address this with a "Crop" alternative and fine-tuning. We also note that CutMix improves spatial relationship learning.
>
> In short, our results reinforce that accessible, academic-scale datasets are a viable path for training powerful and reproducible text-to-image models.
>
> ## Addressing weaknesses:
>
> ### More data seems to reach better absolute performance
>
> We agree that large-scale commercial models such as _SD3_ and _Janus_ currently achieve the highest absolute performance, and we do not claim to outperform them. Our goal is not to replace such models, but rather to **challenge** the prevailing assumption that only massive datasets and models can yield strong results.
>
> Our work demonstrates that with **reproducible**, **resource-efficient training** on a small and well-known dataset like **ImageNet**, it is still possible to **achieve competitive performance**.
>
> #### Regarding data efficiency:
>
> To make this comparison more explicit, we introduce a **data-efficiency-aware metric**, defined as:
>
> $$\text{score}_{\text{eff}} = \frac{\text{score}}{\log(\text{dataset size})}$$
>
> This metric accounts for both performance and dataset size. As shown in Table D.1, our models outperform many large-scale baselines on this adjusted score, reinforcing our central claim: **smaller datasets and models can yield excellent performance per unit of data**.
>
> #### Table D.1: Comparison with SOTA models on DPG Bench adjusted w.r.t $\text{score}_\text{eff}$
> | Model | Params | Training set size | Global | Entity | Attribute | Relation | Other | Overall |
> | --- | --- | --- | --- | --- | --- | --- | --- | --- |
> | Pixart-$\Sigma$ ($512\times 512$)              | 0.6    | 35M               | 5.04   | 5.01   | 4.98      | 4.84     | 4.96  | 4.58    |
> | Pixart-$\Sigma$ ($1024\times 1024$)            | 0.6    | 35M               | 5.00   | 4.77   | 5.12      | 4.99     | 5.05  | 4.63    |
> | Sana-0.6B                   | 0.6    | 1B+               | 3.99   | 4.34   | 4.28      | 4.35     | 4.43  | 4.07    |
> | SDXL                        | 3.5    | 5B+               | 3.73   | 3.69   | 3.62      | 3.88     | 3.60  | 3.34    |
> | SD3-Medium                  | 2.0    | 1B+               | 4.24   | 4.39   | 4.29      | 3.89     | 4.28  | 4.06    |
> | Janus                       | 1.3    | 1B+               | 3.97   | 4.22   | 4.23      | 4.12     | 4.17  | 3.84    |
> | DiT-I $256^2$ CutMix (Ours) | 0.4    | 1.2M              | **5.86**   | 6.12   | 6.04      | 6.53     | 5.34  | 5.54    |
> | CAD-I $256^2$ CutMix (Ours) | 0.3    | 1.2M              | 5.78   | **6.25**   | **6.10**      | **6.68**     | **5.57**  | **5.71**    |
> | DiT-I $512^2$ (Ours)        | 0.4    | 1.2M              | 5.71   | 5.94   | 5.96      | 6.44     | 5.14  | 5.37    |
>
>
>
> ### What about using another data-source than Imagenet with the same amount of data?
>
> To evaluate the generalizability and robustness of our approach, we trained models on **other small-scale datasets** using the same pipeline (captioning, augmentation, architecture, and compute budget). Specifically, we experimented with:
>
> - ~$1$M samples from _DataComp_
>
> - $470$k images from _LAION-POP_ (as suggested by Reviewer tjpV)
>
> - _ImageNet_ combined with $120$k _COCO_ images to reduce object-centric bias
>
> The GenEval results are shown in Tables D.2 (copied from Table C.1 in the answer to Reviewer he9B) and D.3:
>
> #### Table D.2 (copied from C.1): Comparison on GenEval Benchmark between different training datasets with the `DiT-I` architecture.
>
> | Dataset         | Overall  | One obj. | Two obj. | Count.   | Col.     | Pos.     | Col. attr. |
> | --------------- | -------- | -------- | -------- | -------- | -------- | -------- | ---------- |
> | ImageNet        | 0.54     | **0.96** | 0.56     | 0.38     | **0.79** | 0.22     | 0.33       |
> | ImageNet + COCO | **0.55** | 0.93     | **0.59** | **0.42** | 0.77     | **0.24** | **0.34**   |
> | LAION-POP       | 0.21     | 0.64     | 0.13     | 0.09     | 0.32     | 0.04     | 0.02       |
>
> #### Table D.3: Comparison on GenEval Benchmark between ImageNet-1.2M and Datacomp-1M, using `CAD-I` architecture
>
> | Dataset     | IA   | Overall | One obj. | Two obj. | Count. | Col. | Pos. | Col. attr. |
> | ----------- | ---- | ------- | -------- | -------- | ------ | ---- | ---- | ---------- |
> | ImageNet    | ×    | 0.55    | 0.97     | 0.60     | 0.42   | 0.74 | 0.26 | 0.35       |
> | ImageNet    | Crop | 0.54    | 0.96     | 0.61     | 0.40   | 0.71 | 0.23 | 0.33       |
> | DataComp-1M | ×    | 0.34    | 0.84     | 0.28     | 0.18   | 0.57 | 0.05 | 0.13       |
> | DataComp-1M | Crop | 0.38    | 0.86     | 0.35     | 0.25   | 0.55 | 0.13 | 0.16       |
>
>
> Key Takeaways:
>
> - _ImageNet+COCO_ **improves compositionality** (e.g., Two Objects and Counting), suggesting that reducing ImageNet’s object-centric bias helps.
>
> - _ImageNet_ **outperforms _LAION-POP_** when used for **training from scratch**, likely due to its more **balanced** and **concept-uniform distribution**.
>
> - _DataComp-1M_ **lags behind significantly** despite similar scale, possibly due to **noise, and e-commerce bias**.
>
> A promising direction for future work is to explore **combinations of high-quality academic datasets** (e.g., ImageNet, COCO, CC12M) to build a **robust** yet **reproducible training set**, extending what we began with ImageNet+COCO.
>
>
> ### Move details about CutMix from Appendix E-1 to the main paper.
>
> Thank you for the suggestion, we will do in the camera ready version.
>
>
> ### Include results showcasing more complex multi-object scenarios to better demonstrate the model’s text-to-image (T2I) generation capabilities.
>
>
> Thank you for the suggestion, we will add a full page of such examples in the camera ready version. Note that **we already have such images in the paper**, for example the **Kangaroo with sun-glasses in front of Sydney's Opera House** in the teaser **depict a complex interaction with 3 concepts** that are probably not seen together in _ImageNet_. Same for the **Corgy with the pizza** or some of the DPGBench prompts in Appendix A.2.
>
> ## Adressing questions:
>
> ### Can we apply this to other datasets?
>
> See the answer to the corresponding weakness.
>
> ### Are there any limitations to the CutMix strategy and how are they addressed?
>
> The main limitation in the CutMix strategy is that some of the captions are easily distinguishable from non-CutMix captions (e.g., “The left part of the image [...]. The right part of the image [...]”). In turns, the model learns these characteristics and has a chance to **produce artifacts** when prompted with a CutMix caption (which is not the case for human made captions). This is the reason for also proposing the **Crop strategy** which does not suffer from such bias. In any case, **fine-tuning without CutMix augmentation gets rid of any artifacts**. This is what we do for higher resolution models (512, 1k).
>
> ### How does the model handle concepts that are rare or absent in ImageNet?
>
>
> This is a question we also found fascinating and that **we have explored in depth since the submission deadline••. We wish we could show images as in the previous NeurIPS instances, but alas. To test this, we tried prompts containing objects that did not exist at the time of ImageNet’s creation such as an iPad or virtual reality goggles like the HoloLens. The images we get are high quality (about the same as for regular prompts), and the new concept is **hallucinated from patterns that are in the training set.** For example, the **iPad** is made from a **shortened CRT monitor** and the **HoloLens** are made from a **scuba-diving mask with plastic inserts on the side**. We believe this is thanks to **the text encoder** that knows about these concepts and produces features that the diffusion model has to interpret as visual structures. It could be an indication that the closest visual structure to a HoloLens is a scuba-diving mask. We will definitely add examples of such cases in the final version.
>
> ### How does CutMix contribute to learning spatial relationships? Could the approach be extended to involve more than two objects?
>
>
> CutMix does **improve** a bit the **two-objects category on GenEval** (see Table 2), which could indicate **better spatial relationship learning**. Merging more than 2 images is definitely possible, but it would then become **tricky** to design a strategy that produces images without harmful artifacts. This is an interesting research direction.

---

> > ### Comment · Reviewer_fyFz · 2025-08-01
> > **Rebuttal Followup**
> >
> > Thank you to the authors for the detailed and thoughtful responses, as well as the additional experiments provided. I really appreciate the effort! After reading through the response, I still have a couple of points I’d like to follow up on:
> >
> > **1. Positioning of the Paper**
> >
> > I understand and appreciate that the main goal here is to achieve competitive results using a small-scale dataset, which is impressive considering how much compute and data go into many state-of-the-art models.
> >
> > That said, from a user or industry perspective, people often tend to go with the strongest-performing models—regardless of how much data they require. So even with promising small-scale training techniques, many teams may still default to training large models on massive datasets.
> >
> > One suggestion: it might strengthen the paper to explicitly compare models along the data scale, say, by aligning them on the same x-axis in Fig. 2. That kind of comparison could really highlight the benefits of your approach. It’s exciting to see that your results might already be forming a Pareto frontier in that figure. If emphasized more clearly, it could be quite compelling.
> >
> > **2. Performance Outside of ImageNet**
> >
> > Looking at Table D.2, the LAION-based results show a significant performance drop compared to ImageNet. Since LAION is widely used for training due to its diverse text content, this result is a bit concerning.
> >
> > The explanation around ImageNet having a more uniform distribution makes sense but does this imply that the proposed pipeline can’t be directly applied to other large-scale datasets without extra data balancing or manipulation?
> >
> > From my perspective, being able to transfer this pipeline to more diverse datasets would be an important step, especially for T2I tasks where a wide range of prompts is common. It’d be great to see whether this approach could generalize out-of-the-box, or if there are concrete suggestions for how to adapt it to more complex datasets.

---

> > > ### Author Response · Authors · 2025-08-01
> > >
> > > We thank the reviewer for their very insightful discussion. As requested, here is the follow up to the points that the reviewer touched upon.
> > >
> > > ## 1. Positioning of the Paper
> > >
> > > Thank you for the suggestion! Regarding comparison on the same data scale, unfortunately most SOTA models do not provide their training procedure and it is thus impossible to retrain these models at smaller scale to analyze data efficiency. We believe that our work is less about data efficiency than about **proposing a reproducible and comparable protocol** for training text-to-image models. Our setup keeps **one variable fixed (data)** so that future work can vary architecture or training strategy and compare **on a transparent benchmark.**
> > >
> > >
> > > From the industry perspective: Another advantage of pre-training on a small but well curated dataset is that **you know what the model has not been trained on**. For example, LAION has been taken down in the past for containing CSAM content, which poses serious concerns about models that have been trained using these data. We show that you can essentially **gain the same general capabilities as large-scale pretraining** without having that risk. Then, **high aesthetics capabilities** or specific styles can always be obtained through **fine-tuning**, as has been shown by the numerous LoRAs existing for various models.
> > >
> > > We also think that even though big players will keep pretraining on their own dataset, **sharing ablations and experiments on this common dataset** can only **benefit** the field and therefore industrial applications down the line.
> > >
> > > **Dataset reproductiblity** is one of the **biggest slowdowns in our field**. One particular example is CLIP. 4 years after the release of CLIP, the original CLIP is still one of the best CLIP models out there and the main **bottleneck is clearly the dataset**. Efforts like MetaCLIP tries to replicate the dataset configuration but still leads to subpar models.
> > >
> > > ## 2. Performance Outside of ImageNet
> > >
> > > There is maybe a misunderstanding: _LAION-POP_ is **only a small subset of _LAION_** that contains ~470K images of high resolution and high aesthetic scores. Because it is **heavily filtered for image quality**, it is also much **more biased** than _ImageNet_. This is why _LAION-POP_ is under-performing as a pre-training dataset compared to _ImageNet_.
> > >
> > > However, when used as a **fine-tuning dataset** (see Table A3 in answers to reviewer tjpV - top of the page), it allows us to train at $1024\times 1024$ resolution and **improve the aesthetics quality** of the model. This fine-tuned model also gains new capabilities like being able to **generate cartoony images** which is difficult by training on _ImageNet_ alone.
> > >
> > > This makes us think that the **pareto optimal strategy** may look like a sequence of **smaller but well curated dataset**: something very broad and well-balanced like _ImageNet_ for the **pretraining**, and something more **specific** (high resolution, high aesthetics for creative content generation, road images for autonomous driving, etc) for **fine-tuning**. We do believe that some sort of curation is indeed necessary for the pretraining phase. Given our experinments with randomly selected 1M samples from Datacomp, the curation intuition becomes more stronger.

---

> > > > ### Comment · Reviewer_fyFz · 2025-08-02
> > > > **Rebuttal Followup**
> > > >
> > > > Thank you for addressing all of my questions. I’ve finalized my rating.
> > > >
> > > > After the discussion, I’d like to offer a couple of suggestions that might be interesting directions for future work:
> > > >
> > > > 1. Exploring how to select an effective subset from large-scale datasets like LAION or DataComp seems like a very promising direction. These datasets contain much richer and more diverse prompts than ImageNet, so combining your proposed approach with a thoughtful selection strategy could really push things forward. It would be great to see if a well-chosen subset can get us close to large-scale performance with much less data.
> > > > 2. The use of CutMix to improve multi-object scenarios is an interesting example. It opens up a broader question: what kinds of data augmentation can help guide model training, especially when working with small-scale datasets? Augmentations that add useful bias (like for text-on-image, high aesthetics, or color richness) could help unlock stronger performance in specific use cases.
> > > >
> > > > Overall, it’s been a pleasure engaging with the ideas in your paper!

---

> > > > > ### Author Response · Authors · 2025-08-02
> > > > >
> > > > > We would like to thank the reviewer for their insightful discussion, prompt answers and shedding light on prominent future directions.

---

### Official Review · Reviewer_he9B · 2025-06-28

**Clarity:** 3
**Significance:** 2
**Originality:** 2
**Rating:** 4
**Confidence:** 4

**Summary:**

The paper challenges the "bigger is better" paradigm in text-to-image generation by demonstrating that models trained solely on ImageNet with well-designed text and image augmentations can match or outperform state-of-the-art models trained on billion-scale datasets. The findings are surprising and can benefit the GenAI research.

**Questions:**

How to further improve multi-object composition without increasing data?

**Ethical Concerns:**

["NO or VERY MINOR ethics concerns only"]

**Final Justification:**

My major concerns have been addressed.

**Limitations:**

Yes

**Quality:**

3

**Strengths And Weaknesses:**

Strengths:
1. By using ImageNet (1.2M images) with synthetic captions and augmentations (e.g., CutMix, Crop), the model achieves good performance.
2. ImageNet’s standardization and open access address critical issues in the field (closed-source data, decay), enabling reproducible research. The training setup requires ~500 H100 hours, making it accessible to smaller teams.
3. Metrics include FID, CLIPScore, GenEval, DPGBench, and aesthetic scores, covering image quality, text-image alignment, and human preference, have been reported.

Weaknesses:
1. The novelty is limited; I mean the idea is novel, while the paper does not propose any new approaches, the key approaches in the paper are re-caption and augmentation. The former one is proposed by DALL-E 3, and the last one has been widely deployed among computer vision tasks.
2. ImageNet’s object-centric bias; The dataset lacks complex scenes (e.g., humans, environments), leading to limitations in generating diverse contexts. This is the limitation of ImageNet, because this is a 1000-class classification dataset. For better visualization, maybe you can have a minor part on discussing how to improve it, by adding more data from a large-scale dataset such as CC3M, JourneyDB, etc.
3. The 512^2 resolution is too small for generation, although I understand generating 512^2 resolution images is very difficult if using Imagenet. Can you use a few number of 1024^2 samples to continue to scale up the resolution?

---

> ### Author Rebuttal · Authors · 2025-07-31
>
> ## TL;DR
>
> We thank the reviewer for their valuable feedback. We have addressed the weaknesses and questions raised by the reviewer, and we summarize our key points and new experimental findings below:
>
> * **Methodological Novelty:** We acknowledge that our core techniques (long captions and image augmentations) are not novel. Our core contribution is demonstrating that a reproducible, resource-efficient training on a small, open dataset like ImageNet can achieve state-of-the-art performance, matching or exceeding models trained on orders of magnitude more data. Our work advocates for a standardized, accessible, and reproducible training paradigm for text-to-image generation.
>
> * **New Experiments:**
>     * **Compositional Capabilities:** We trained a new model on a combined ImageNet+COCO dataset to test for object-centric bias. The results show a slight improvement on multi-object compositional tasks like counting and two-object generation, suggesting that while our augmentation strategy is effective, having a more diverse dataset still offers benefits.
>     * **Scaling to Higher Resolutions:** We fine-tuned our model to $1024 \times 1024$ resolution on both a high-resolution ImageNet subset and LAION-POP. Both models successfully generate high-quality images. The LAION-POP model showed a noticeable improvement in aesthetic scores, suggesting that ImageNet may have limitations for scaling to higher resolutions due to its lack of diverse high-res content.
>     * **Improving Multi-Object Composition:** We believe that further improving multi-object composition is a crucial open question. While our current approach helps, a dedicated research direction could focus on developing cost-effective data augmentation strategies that combine objects and generate appropriate captions without relying on large, noisy datasets. This is a very interesting direction for future work.
>
> ## Adressing Weaknesses:
>
> ### Novel idea but lack of methodogical novelty
>
> Indeed, we fully acknowledge that using long captions and image augmentations is not novel, and we do not claim these techniques as our contribution.
>
> The central goal of our paper is to challenge the **“bigger is better”** paradigm by showing that **reproducible, resource-efficient training** on **smaller datasets, specifically ImageNet**, can still yield strong performance; comparable to state-of-the-art models trained on **orders of magnitude more data**.
>
> Our main contribution is to demonstrate that it is possible to:
>
> - Train models from scratch on a small, open, and well-known dataset (ImageNet);
>
> - Avoid reliance on __massive web-scraped datasets__;
>
> - Match or exceed the performance of popular models like SDXL on key compositional benchmarks;
>
> - All while using orders of magnitude less data and compute.
>
> Our intent is to advocate for a **standardized, accessible, and reproducible training setup for text-to-image generation**. We believe that promoting reproducibility, accessibility, and sustainability in generative model research (in particular text-to-image generation) is important and timely message for the community.
>
>
>
> ### Can we improve on Imagenet object centric bias?
>
>
> We train two additional models: one on _ImageNet$+$COCO_ and one on _LAION-POP_ (as suggested by Reviewer tjpV) to assess whether having a dataset without the object centric bias helps. The GenEval results are:
>
> #### Table C.1: Comparison on GenEval Benchmark between different training datasets with the `DiT-I` architecture.
> Dataset | Overall| One obj.| Two obj. | Count. | Col.| Pos.| Col. attr.|
> | --- | --- | --- | --- | --- | --- | --- | ---|
> | ImageNet ($1.2$M)|0.54|**0.96**|0.56|0.38|**0.79**|0.22|0.33|
> | ImageNet ($1.2$M)$+$COCO ($120$k) | **0.55** | 0.93 | **0.59** | **0.42** | 0.77 | **0.24** | **0.34** |
> | LAION-POP ($470$k)| 0.21 | 0.64 | 0.13 | 0.09 | 0.32 | 0.04 | 0.02 |
>
>
> As we can see, adding _COCO_ to the training mix does not change the overall results very much, which indicates that the augmentation strategies are enough to get general capabilities. The model trained with _COCO_ performs better on the Two Objects task (0.59 vs. 0.56) and Counting (0.42 vs. 0.38), suggesting that reducing the object-centric bias of _ImageNet_ helps improve multi-object compositional reasoning.
>
> _LAION-POP_ is either too small or not diverse enough to lead to satifying results. More advanced capabilities (like high quality faces or aesthetics images) could be acquired after pretraining on _ImageNet_ by fine-tuning on a dedicated dataset (see Table A.3 in the answer to Reveiwer tjpV)
>
> ### Can we scale further than 512x512 resolution?
>
> To explore higher resolutions, we **fine-tuned** two models at $1024\times1024$ resolution:
>
> - One on a **high-resolution subset of ImageNet**, and
>
> - One on _**LAION-POP**_, as suggested by Reviewer tjpV.
>
> Both models successfully generate high-quality $1024\times1024$ images, with distinct biases: the _ImageNet_ model tends to produce photographic outputs, while the _LAION-POP_ model exhibits a stronger aesthetic bias.
>
> As shown in Table C.2 (copied from Table A.3 in the answer to Reviewer tjpV), the ImageNet-1024 model achieves aesthetic scores comparable to its $512\times 512$ counterpart, while the _LAION-POP-1024_ model shows a noticeable improvement in aesthetic metrics. This suggests a potential limitation of ImageNet for high-resolution generation: its relative lack of high-res content may hinder performance scaling at $1024\times 1024$ and beyond.
>
>
> #### Table C.2 (copied from A.3): Comparison on Aesthetic metrics between ImageNet and LAION-POP for **finetuning** at $1024\times 1024$ . We start our finetuning from the previously reported `DiT-I` Crop $512\times 512$ resolution checkpoint. PickScore and Aesthetic Scores are based on images generated using `Partiprompts` whereas `HPSv2.1` and `ImageReward` are calculated using the respective benchmark prompts.
>
> | Dataset | Resolution | PickScore | Aes. Score | HPSv2.1 | ImageReward |
> |----|---------|-----------|------------|---------|-------------|
> | ImageNet ($1.2$M) | $512\times 512$ | 20.94 | 5.46 | 0.24 | 0.20 |
> | Finetuning on ImageNet ($26$k) | $1024\times 1024$ | 20.36 | 4.96 | 0.22 | -0.42 |
> | Finetuning on LAION-POP ($375$k) | $1024\times 1024$ | **21.04** | **5.67** | **0.25** | **0.24** |
>
> We see this as an interesting direction for future work, particularly in combining data-efficient training with selective use of high-resolution content.
>
>
> ## Addressing questions:
>
> ### How to further improve multi-object composition without increasing data?
>
>
> This is a very interesting open question. We show that mixing images from the training set can improve two object composition (see GenEval score in Table 2). This strategy could be explored by mixing more objects. However, this requires generating captions for the specific mix of objects, and it also requires designing a strategy that avoids creating bad images (cutting objects, floating objects, nonsensical mixes, etc). Solving multi-object composition augmentations without a high caption generation cost is a very interesting future research direction. This would be part of an entire field of research that could be built on top of our proposition of imagenet-based pretraining of T2I models, with fair, reproducible and accessible research.

---

> > ### Comment · Reviewer_he9B · 2025-08-04
> >
> > Thanks for your detailed response! Most of my concerns have been addressed. Good luck!

---

### Official Review · Reviewer_qDs6 · 2025-07-01

**Clarity:** 3
**Significance:** 3
**Originality:** 3
**Rating:** 4
**Confidence:** 4

**Summary:**

This paper aims to explore the potential of training text-to-image (T2I) models using only the ImageNet dataset, in an attempt to challenge the prevailing "bigger is better" paradigm that relies on massive, often private, web-crawled data. The authors propose a training pipeline combining synthetic captions (text augmentation) and specific image augmentations (CutMix, Crop). Through this pipeline, they claim that their lightweight model (~400M parameters) can match or even surpass SOTA models like SDXL, which are orders of magnitude larger in both parameter count and training data size, on specific compositionality benchmarks (e.g., GenEval). However, the paper's core premise, methodological purity, and the comprehensiveness of its evaluation are deeply flawed, leading to a misleading conclusion.

**Questions:**

My questions are listed above in the weaknesses section. If the author could address my concern, I would prefer to raise the final score.

**Ethical Concerns:**

["NO or VERY MINOR ethics concerns only"]

**Final Justification:**

I have read through all the reviews and author responses. I have discussed with the author during the phase and reached a final conclusion based on the rebuttal and discussion. The authors provide comprehensive explanations that have addressed all my main concerns. The new insight of CutMix reinforces the quality and contribution of the paper. However, due to the initial writing quality and the easy data augmentation without enough qualitative examples, the holistic presentation and experiment are somewhat limited. Thus, I raise the final rating from reject to borderline accept, which means that the paper is about ready for publication in the conference.

**Limitations:**

yes

**Paper Formatting Concerns:**

No formatting concerns.

**Quality:**

2

**Strengths And Weaknesses:**

**Strength**

The motivation of the paper is clear, and the author provides two solutions, TA and IA, to close the performance inferior of training only on the ImageNet dataset for the T2V task. The methods are simple and achieve a reasonable gain.

**Weaknessess**
1. The writing needs to be improved. There are too many typos in this paper. The gray color highlight in the Tables is disordered. The nominations, such as "DINOv2," have different font styles.

2. The anonymous link provided by the author is empty, which cannot support reviewing the paper.

3. The motivation of the proposed methods is clear and reasonable. However, the solutions are quite simple and lack novelty.

4. The baselines SD v2.1 and SDXL are not the SOTA method in the current era. Modern baselines such as SD3 and FLUX are required.

5. The effectiveness of the proposed methods is not robust across different models. For example, the CutMix is not effective on CAD-I as in Table 3. More fundamental methods need to be evaluated, such as SiT and traditional U-Net.

6. Why are there no experiments for Crop on CAD-I? Does it achieve unsatisfactory performance or fail to converge?

7. Training only on ImageNet may result in an overfitting problem during the longer training process, which could be found in Figure 3. Thus, in many T2V tasks, a larger training dataset is required to provide more diverse visual patterns and image-text alignment datapoints.

---

> ### Author Rebuttal · Authors · 2025-07-31
>
> ### TL;DR
>
> We appreciate the reviewers' comments and have significantly updated our submission.
>
> **Key Updates & Contributions:**
>
> * **Public Release:** All code and data will be made public upon acceptance.
> * **Simplicity & Reproducibility:** Our core contribution is a **reproducible and accessible framework** for T2I research. We demonstrate that **impressive results are achievable with simple setups** (ImageNet, straightforward methods), challenging the "bigger is better" paradigm and lowering the entry barrier.
> * **New SOTA Baselines:** We've added comprehensive comparisons against **PixArt-$\Sigma$, Janus, FLUX, SANA, and SD3** on GenEval and DPGBench (Tables B.1, B.2 from reviewer tjpV).
>     * Our models **outperform PixArt-$\Sigma$** on GenEval at $512\times 512$ resolution, particularly in compositional tasks (Two Objects, Position, Color Attribution).
>     * Though lagging behind large-scale commercial models (SANA, SD3, FLUX), our models achieve competitive results with **$1000\times$ fewer images** and **$10\times$ fewer parameters** than SDXL.
> * **CutMix Effectiveness:** While not always improving FID, CutMix **significantly enhances compositional understanding** (e.g., Two Objects, Position scores in Table 2) and **mitigates overfitting**, leading to improved training stability (Figure 3).
> * **Robustness Across Paradigms:** We demonstrate the generalizability of our training setup by evaluating **Flow Matching models** (Table B.3). They achieve competitive results comparable to PixArt-$\Sigma$ and SD v2.1, proving our approach is robust across different generative modeling paradigms.
> * **New Crop Augmentation Experiment:** We added experiments with the **Crop augmentation strategy on CAD-I** to complete our ablation studies (Table B.4).
> * **Overfitting Mitigation:** Our augmentations (especially CutMix) effectively **delay overfitting** on ImageNet, enabling strong performance with significantly less data and compute than state-of-the-art models.
>
> # Addressing Reviewer Feedback
>
> We appreciate your thorough review and valuable comments. We've addressed your points below.
>
> ### 1. Typos and Style Discrepancies
>
> We apologize for the typos and inconsistencies in style. We will meticulously proofread and correct these in the camera-ready version of the paper.
>
> ### 2. Broken Code and Data Links
>
> We understand your frustration regarding the non-functional links. These links were placeholders, included to estimate space requirements and **maintain anonymity** during the review process. All code and data are prepared and will be made publicly available upon the paper's acceptance.
>
> ###  3. Simplicity of the Solution
>
> We appreciate your feedback on the simplicity of our solution. Our primary contribution lies in offering the community a **reproducible and easily accessible framework** for Text-to-Image (T2I) research. The simplicity of our approach is, in fact, a deliberate design choice rather than a limitation.
>
> We aim to demonstrate that even with a straightforward dataset like __ImageNet__ and a simple setup, it's possible to achieve **impressive results** that might otherwise be **considered unfeasible**. While more complex methods could potentially yield incrementally better performance, we believe our current setup effectively **highlights the value and potential of this research direction**. Our work underscores that significant insights can be gained without resorting to overly intricate solutions, thereby lowering the **barrier to entry** for other researchers.
>
> ### 4. More Modern baselines
>
> Thank you for the helpful suggestion.
>
> We have updated Tables 5 (GenEval) and 6 (DPGBench) in the main paper to include comparisons with more recent models, including _PixArt-$\Sigma$_, _Janus_, _FLUX_, _SANA_, and _SD3_. Please refer to **Tables B.1** and **B.2** from reviewer tjpV for a preview.
>
> At $512\times 512$ resolution, our models outperform _PixArt-$\Sigma$_ on the GenEval benchmark, particularly in the **Two Objects**, **Position**, and **Color Attribution** sub-tasks. However, we acknowledge that our models still fall short of the strongest commercial baselines such as _SANA_ and _SD3_.
>
> On the DPGBench benchmark, our best model again achieves comparable performances to _PixArt-$\Sigma$_ at $512\times 512$ resolution and _Janus_, but lag behind _SANA_, _SD3_ and _FLUX_, which use larger-scale training and commercial infrastructure.
>
> ### 5. The effectiveness of the proposed methods is not robust across different models. For example, the CutMix is not effective on CAD-I as in Table 3.
>
>
> We acknowledge that CutMix does not consistently improve FID metrics, particularly for the CAD-I model in Table 3. However, we believe its contribution remains essential for two key reasons:
>
> 1. #### Improved Compositionality:
> While FID remains similar, CutMix significantly enhances **compositional understanding**. For example, in Table 2, the **Two Objects** score for CAD-I improves from **0.60** to **0.68**, and the **Position** score improves from **0.26** to **0.35** with CutMix. These gains indicate stronger multi-object reasoning and spatial understanding.
>
> 2. ####  Overfitting Mitigation:
>  As shown in Figure 3, **CutMix helps delay overfitting** during training. The FID curve stays lower for longer, and GenEval scores degrade more slowly, demonstrating improved training stability and generalization under limited data.
>
>
> ### 6. More fundamental methods need to be evaluated, such as SiT and traditional U-Net.
>
> Thank you for the suggestion.
>
> In response, following **SiT**, we trained and evaluated **Flow Matching** models under the exact same conditions as our diffusion-based models. These models use the **CAD-I** architecture, and were trained with identical datasets, compute budgets, and training procedures.
>
> As shown in Table B.3, while Flow Matching models slightly lag behind diffusion-based models in overall performance due to undertraining, they still achieve competitive results, **comparable to _PixArt-$\Sigma$_** (0.52) and _**Stable Diffusion v2.1**_ (0.50) on the **GenEval benchmark**. This demonstrates that our proposed training setup, based on ImageNet with long captions and augmentations, is robust and generalizable across **different generative modeling paradigms.** We believe with more training time and better hyperparameters, the flow matching models could reach or even surpass the diffusion models.
>
> #### Table B.3: Comparison between Flow-Matching and Diffusion models, using the `CAD-I` architecture at $256\times 256$ resolution.
>
> | Generative Paradigm | IA       | Overall | One obj. | Two obj. | Count. | Col. | Pos. | Col. attr. |
> | ------------------- | -------- | ------- | -------- | -------- | ------ | ---- | ---- | ---------- |
> | Diffusion           | $\times$ | 0.55    | 0.97     | 0.60     | 0.42   | 0.74 | 0.26 | 0.35       |
> | Diffusion           | CutMix   | 0.57    | 0.94     | 0.68     | 0.40   | 0.70 | 0.35 | 0.36       |
> | Flow Matching                    |  $\times$        | 0.51         | 0.92         | 0.56          | 0.35        | 0.68      | 0.23     | 0.27           |
> | Flow Matching                    | CutMix         | 0.50         | 0.89          | 0.53          | 0.40        | 0.64      | 0.22     | 0.29           |
>
>
>
> ### 7. Why are there no experiments for Crop on CAD-I? Does it achieve unsatisfactory performance or fail to converge?
>
>
> We originally just did not train with the crop strategy on CAD-I because the CutMix strategy seemed to be sufficient to get good performances.
>
> For the rebuttal, we have trained a CAD-I model with the Crop augmentation strategy to complete the ablation. The training followed the same setup as our other CAD-I experiments.
>
> Table B.4 contains the results that are in line with the ones obtained for DiT-I.
>
> #### Table B.4: Comparison between Image Augmentation strategies with the `CAD-I` architecture.
> | Model | IA     | Overall | One obj. | Two obj. | Count. | Col. | Pos. | Col. attr. |
> | ----- | ------ | ------- | -------- | -------- | ------ | ---- | ---- | ---------- |
> |        | ×      |  0.55  | 0.97  | 0.60 | 0.42 | 0.74 | 0.26 | 0.35 |
> | CAD-I  | Crop   |  0.54  | 0.96  | 0.61 | 0.40 | 0.71 | 0.23 | 0.33 |
> |        | CutMix |  0.57  | 0.94  | 0.68 | 0.40 | 0.70 | 0.35 | 0.36 |
>
>
> ### 8. Training only on ImageNet may result in an overfitting problem during the longer training process, which could be found in Figure 3. Thus, in many T2V tasks, a larger training dataset is required to provide more diverse visual patterns and image-text alignment datapoints.
>
> We agree that overfitting is a common concern when training on small datasets like ImageNet. However, as shown in Figure 3, our proposed strategy, particularly the use of image augmentations such as CutMix, significantly mitigates overfitting over the course of training. The FID remains stable for longer, and the GenEval score degrades more slowly.
>
> Our approach does not require prolonged training to reach competitive performance. As demonstrated in Tables B.1 and B.2 from reviewer tjpV, our model already matches _SDXL_ on GenEval while being trained with **$1000\times$ fewer** images,  **$10\times$ fewer** parameters and **a fraction of the compute budget**.
>
> This supports the central goal of our paper: to **challenge the "bigger is better"** paradigm by showing that with thoughtful data augmentation and captioning strategies, **reproducible, resource-efficient** training on **smaller datasets** can still yield strong performance. Exploring how to extend training duration or scale model capacity further while maintaining data efficiency is indeed an interesting direction, and we leave this for future work.

---

> > ### Comment · Reviewer_qDs6 · 2025-08-02
> >
> > Thank you for the detailed and thoughtful response.
> >
> > I appreciate the additional comparisons against recent baselines (PixArt-Σ, FLUX, SD3, SANA, Janus), the backbone compatible experiments, and the completed CAD-I + Crop ablation. These additions directly address my original concerns.
> >
> > I also respect the deliberate emphasis on simplicity and reproducibility; in resource-constrained settings, a lightweight pipeline that still delivers strong results is indeed valuable. By demonstrating that competitive T2I performance is achievable with only ImageNet and modest compute, you lower the entry barrier for academic groups and enable rapid iteration on new ideas without the burden of billion-scale datasets. This is particularly important for fostering open science and for domain-specific adaptation where large-scale data collection is impractical.
> >
> > One minor suggestion for the camera-ready version: a short qualitative analysis—e.g., illustrating why CutMix specifically improves multi-object or positional understanding—would further strengthen the paper. The typos need to be fixed, and the rebuttal discussion should be included in the revised version. I look forward to the public release of the code and data. I am inclined to raise the final score.

---

> > > ### Author Response · Authors · 2025-08-02
> > >
> > > We thank the reviewer for participating in the discussion and we are really happy to hear that the reviewer finds our simple method highly beneficial for the community. We will correct all typo errors and incorporate the feedback from the rebuttal into the final camera-ready version of the paper.
> > >
> > > ## Short qualitative analysis of CutMix
> > > We thank the reviewer for refering us to point out the qualitative analysis on why CutMix seems to be better in compositionality. We believe that this **improvement is majorly credited to the recaptioning of the cutmix images**. The captions of cutmix images tend to include very **rich details about positioning and multi-concept attributions**. We do provide examples of cutmix images and their corresponding captions in **Figure 7** of appendix. The additional information really helps the model to gain more insight about **compositionality** (see comparison of `a pirate ship sailing on a steaming soup` in Figure 4) leading to much better scores in Geneval and DPG benchmarks. To further highlight this hypothesis, we will include additional qualitative examples comparing generations from models trained with and without CutMix, specifically on compositional prompts. For ease of readability we will add reference to the Figure 7 in the main paper for the camera ready version.
> > >
> > > We also want to point out that as mentioned by reviewer fyFz, augmentations like CutMix open a new research avenue about which images or text augmentations are the most effective at increasing compositionality from a limited set of starting images.

---

> > > > ### Comment · Reviewer_qDs6 · 2025-08-03
> > > >
> > > > Thank you for the follow-up discussion. I am aware of the qualitative analysis of CutMix in Figure 4 and the data examples in Figure 7. My suggestion is to provide more ablations, as in Figure 4, demonstrating the compositionality brought by CutMix. The augmentation is novel to the field, and I hope this avenue grows with more novel and insightful research in the future. The final score is raised. Good Luck!

---

### Official Review · Reviewer_tjpV · 2025-07-02

**Clarity:** 2
**Significance:** 2
**Originality:** 2
**Rating:** 2
**Confidence:** 5

**Summary:**

This paper challenges the prevailing belief that massive web-scale datasets (e.g., LAION-5B) are essential for high-quality text-to-image (T2I) generation. The authors propose a simple yet effective alternative: training T2I diffusion models solely on ImageNet, augmented with synthetic long captions and compositional image augmentations. They adapt the DiT and CAD architectures for this setup and demonstrate that models trained on only 1.2M images (i.e., ImageNet) can outperform or match large models like SDXL on GenEval and DPG-Bench using just 1/1000th of the data and 1/10th the parameters. Their approach emphasizes reproducibility, efficiency, and interpretability through open data and code.

**Questions:**

1. What resolution was used when generating images with SDXL for comparison? The SDXL samples in the paper (e.g., Figure 5) appear to be of relatively low quality, which raises concerns that the model may have been run at 512×512 resolution. Since SDXL is trained and typically evaluated at 1024×1024, using a lower resolution could significantly degrade output quality and lead to an unfair comparison.

2. Comparison with Recent State-of-the-Art T2I Models:
The paper compares its models against SDXL and PixArt-α, which are relatively strong baselines. However, both of these models do not utilize enriched captions during training, which may explain their weaker performance on compositionality-focused benchmarks like DPGBench. Could the authors provide comparisons against more recent models that do use enriched or multi-level captions, such as PixArt-Σ (Sigma), SANA, or FLUX? Including these would provide a more thorough and up-to-date evaluation.

**Ethical Concerns:**

["NO or VERY MINOR ethics concerns only"]

**Final Justification:**

After reviewing the authors’ rebuttal and considering the discussion, I have decided to maintain my original evaluation. While the paper presents an interesting approach, several of my core concerns remain only partially addressed.

- Realism and dataset characterization:
The rebuttal states that models trained on artistic datasets are unable to generate realistic images for certain domains. However, datasets such as LAION-POP and LAION-Aesthetic are not limited to visually artistic or stylized content; they also contain a large number of high-quality, photo-realistic images, in many cases more so than ImageNet. Furthermore, aesthetic metrics such as HPSv2 and ImageReward do not merely capture visual beauty but reflect human preference for overall image quality, which includes realism. Therefore, I am not fully convinced that the claim regarding the inability to generate realistic images is well supported.

- Compositionality comparison:
The observed compositionality improvement for the proposed approach appears to be largely due to text augmentation via an image captioner, rather than the ImageNet dataset itself. This directly challenges the claim that the method outperforms SDXL in compositionality, as SDXL is an older model (2023) that was not trained with enriched captions, whereas the proposed approach benefits from them. Moreover, Table B.2 shows that the proposed method underperforms PixArt-Sigma—a more recent text-augmented model—on compositionality metrics, suggesting the claimed advantage does not extend to stronger baselines.

- Overall impact:
Given the existence of large-scale datasets like LAION, which already provide higher resolution, broader sample diversity, and enriched captions, it is unclear how much additional impact ImageNet with augmented text captions will have on advancing text-to-image generation in practice. While the work demonstrates some promising results, its broader significance appears more limited than suggested.

For these reasons, while I acknowledge the authors’ clarifications and additional explanations, the rebuttal does not fully resolve my main concerns, and I am maintaining my original score.

**Limitations:**

Yes

**Quality:**

2

**Strengths And Weaknesses:**

# Strengh

### Reproducibility & Simplicity:
Uses the widely available and standardized ImageNet dataset, making the approach reproducible and accessible without relying on proprietary or decaying datasets.

### Strong Efficiency:
Achieves competitive or superior performance (e.g., +1% on GenEval, +0.5% on DPG-Bench) compared to state-of-the-art models with drastically less compute, data, and model size.

### Well-Analyzed Augmentation Strategy:
Carefully investigates and validates the effectiveness of both textual (LLaVA-based long captions) and visual (e.g., CutMix, cropping) augmentations to overcome limitations of ImageNet.

### Compositional Strength:
Despite the limited dataset, the proposed models exhibit impressive compositional generalization and prompt following, demonstrated both quantitatively and qualitatively.

# Weakness

### Lack of Comparison with High-quality (high aesthetic & high resolution & enriched caption) Dataset - LAION-POP [1]:
The paper does not compare its approach to models trained on high-quality datasets such as LAION-POP, which consists of high-aesthetic, high-resolution images with enriched captions generated by strong vision-language models like LLaVA. Unlike ImageNet—which primarily consists of low-resolution (≤512×512), object-centric, and realistic photos—LAION-POP supports both low- and high-resolution model training due to its consistently high-quality images. Recent work such as the KOALA paper [2] has demonstrated that, despite its relatively small size (~600K images), LAION-POP enables models to outperform those trained on larger-scale datasets (e.g., 2M images) in terms of visual quality and compositional alignment. Furthermore, in Line 254, the authors argue that their findings highlight the potential of smaller datasets to achieve state-of-the-art results, thereby enabling specialized domain adaptation where large-scale data collection is impractical. However, this claim is not entirely novel. The KOALA study has already shown that LAION-POP—a smaller but purpose-built dataset for text-to-image generation—can lead to competitive or even superior performance. This precedent challenges the necessity of using ImageNet for training text-to-image models and calls into question the novelty of using smaller datasets as a contribution. Without a direct comparison or justification for choosing ImageNet over more suitable datasets like LAION-POP, the generality and practical significance of the authors’ findings remain unclear.

### Limited Evaluation of Aesthetic Quality:
While the paper does include some quantitative aesthetic evaluation in Table 4 using metrics like PickScore, Aesthetic Score, and VILA, these automated measures alone are not sufficient to assess the nuanced and subjective nature of visual aesthetics in text-to-image generation. In particular, human perception of beauty, style, and realism often diverges from what these metrics capture. The absence of human evaluation therefore remains a critical limitation, especially given that aesthetic quality is a key criterion for practical T2I applications. Furthermore, the paper could be improved by incorporating more reliable automated metrics for aesthetic assessment, such as HPS (Human Preference Score) [1] and ImageReward [2], which have been shown to better align with human judgments. A human study—complemented by these stronger metrics—would provide a more comprehensive and interpretable evaluation of aesthetic quality.

### Limited Practical Quality of Generated Images and qualitative comparison with SoTA  open-source models:
While the authors claim that “smaller datasets can achieve state-of-the-art results,” the qualitative results presented in Figures 1, 4, and 6 suggest otherwise. Many of the generated images appear to lack fine-grained detail, visual coherence, or aesthetic appeal. This raises concerns about the practical usability of the proposed approach in real-world scenarios such as creative content generation, user-facing applications, or commercial design tools—domains where high visual quality is often a non-negotiable requirement. Moreover, the paper does not provide qualitative comparisons with recent state-of-the-art open-source text-to-image models such as KOALA, PixArt-series, or SANA-series. These models have demonstrated strong visual fidelity and compositional accuracy, even in challenging prompts. Without side-by-side visual comparisons, it is difficult to assess the relative strengths and limitations of the proposed method, especially given its reliance on ImageNet—a dataset not originally intended for generative tasks.

Including such comparisons would help clarify the practical trade-offs between efficiency and quality and strengthen the authors’ claims regarding the viability of training high-performing generative models from smaller or more accessible datasets.

---
[1] https://laion.ai/blog/laion-pop/
[2] Lee et al. KOALA: Empirical Lessons Toward Memory-Efficient and Fast Diffusion Models for Text-to-Image Synthesis. NeurIPS 2024.
[3] Wu et al. Human Preference Score v2: A Solid Benchmark for Evaluating Human Preferences of Text-to-Image Synthesis. Arxiv 2024.
[4] Xu et al. ImageReward: Learning and Evaluating Human Preferences for Text-to-Image Generation. NeurIPS 2023.

---

> ### Author Rebuttal · Authors · 2025-07-31
>
> ## TL;DR: ImageNet Outperforms LAION-POP & High-Res Capabilities Demonstrated
>
> **New experiments confirm:**
> * **ImageNet (trained from scratch):** Consistently **outperforms LAION-POP** on compositional understanding (GenEval) and even aesthetic metrics. ImageNet offers better generalization and reproducibility.
> * **LAION-POP:** While fine-tuning on it improves aesthetics, training from scratch yields **significantly worse compositionality**. Issues with dead URLs and inappropriate content also exist.
> * **High-Resolution:** Successfully **fine-tuned our model to $1024\times 1024$** on high-res ImageNet, achieving comparable aesthetic quality.
> * **Competitiveness:** Our ImageNet-trained models (smaller, less data) show **competitive performance** on benchmarks (GenEval, DPGBench) against recent SOTA models like PixArt-$\Sigma$.
>
> **Conclusion:** ImageNet remains a strong, reproducible dataset for competitive diffusion model training, especially for compositional understanding.
>
> # Addressing Weaknesses:
>
> ### Lack of Comparison with High-quality (high aesthetic & high resolution & enriched caption) Dataset - LAION-POP [1]:
>
> We appreciate the reviewer’s detailed feedback and the reference to _LAION-POP_ and the _KOALA_ study.
>
> We conducted additional experiments:
> - We trained a model **from scratch** on _LAION-POP_ (only ~$470$k images because of url decay over time) at $256\times 256$ resolution in Tables A.1 and A.2
> - We **fine-tuned** our _ImageNet_ trained model at $1024\times 1024$ resolution using $1024$ resolution _LAION-POP_ samples (~$375$k images) in Table A.3
> - We **fine-tuned** a $1024\times 1024$ resolution model on high-resolution ImageNet samples (only ~$26$k images) in Table A.3
>
> #### Table A.1: Comparison on GenEval Benchmark between ImageNet and _LAION-POP_ for our `DiT-I` model trained __from scratch__ at $256\times 256$ resolution.
> | IA   | Dataset   | Overall | One obj. | Two obj. | Count. | Col. | Pos. | Col. attr. |
> | --- | ---| ---| ---| ---| ---| ---| ---| ---|
> | Crop | ImageNet ($1.2$M) | **0.54** | **0.96** | **0.56**| **0.38**| **0.79**| **0.22** | **0.33**|
> | Crop |  LAION-POP ($470$k) | 0.21| 0.64| 0.13| 0.09| 0.32 | 0.04 | 0.02      |
>
> #### Table A.2: Comparison on Aesthetic metrics between ImageNet and _LAION-POP_ for our `DiT-I` model trained __from scratch__ at $256\times 256$ resolution. Scores are calculated using `PartiPrompts`.
> | IA | Dataset | PickScore | Aes. Score | HPSv2.1 | ImageReward |
> |---|---|---|---|---|---|
> | Crop | ImageNet ($1.2$M) | **20.63** | **5.17** | **0.24** | **0.12** |
> | Crop | LAION-POP ($470$k) | 19.50 | 4.17 | 0.19 | -0.98 |
>
> #### Table A.3: Comparison on Aesthetic metrics between ImageNet and LAION-POP for **finetuning** at $1024\times 1024$, starting from the `DiT-I` Crop $512\times 512$ resolution checkpoint. PickScore and Aesthetic Scores are using `Partiprompts` whereas `HPSv2.1` and `ImageReward` are calculated using the respective benchmark prompts.
>
> | Dataset | Resolution | PickScore | Aes. Score | HPSv2.1 | ImageReward |
> |---|---|---|---|---|---|
> |ImageNet ($1.2$M) | $512\times 512$ | 20.94 | 5.46 | 0.24 | 0.20 |
> |Finetuning on ImageNet ($26$k) | $1024\times 1024$ | 20.36 | 4.96 | 0.22 | -0.42 |
> |Finetuning on LAION-POP ($375$k) | $1024\times 1024$ | **21.04** | **5.67** |**0.25**|**0.24**|
>
> ### Results
> **Fine-tuning** on _LAION-POP_ **improves aesthetic metrics**, aligning with the KOALA findings. However, our main motivation is to train a model from scratch. As such when **training from scratch** on _LAION-POP_, we observe **significantly worse compositionality**. In contrast, ImageNet-based models exhibited **better generalization and compositional understanding** under the same training conditions.
>
> ### ImageNet vs LAION-POP:
> We intentionally chose ImageNet for its **accessibility, reproducibility, and stable long-term hosting**. While _LAION-POP_ offers high-quality images, it suffers from **reproducibility issues**. Over 20% of the image urls are **already unavailable** on HuggingFace. We could only retrieve ~$470$k out of $600$k images. Additionally, _LAION-POP_ contains **inappropriate content** (e.g., NSFW material appears within the first few hundred samples from HuggingFace).
>
> As shown in Table A.1, **ImageNet-trained models outperformed those trained on LAION-POP in both aesthetic metrics quality** and **GenEval**.
>
> ### Comparison with KOALA:
>
> We believe there may have been a misunderstanding: _KOALA_ uses _LAION-POP_ in a fine-tuning/distillation setup **starting from an SDXL checkpoint** that has already been trained on **over 1B images**. Our work is different: **we train models from scratch using only ~1M images**. Models trained with  _KOALA_ benefit from all the images seen during the _SDXL_ pretraining, which is crucial as shown in Table A.1.
>
> ### Limited Evaluation of Aesthetic Quality:
>
> We have added the ImageReward and HPSv2 as metrics in the comparison tables.
>
> ### Limited Practical Quality of Generated Images and qualitative comparison with SoTA open-source models:
>
> We acknowledge that for certain applications, particularly in creative industries or commercial design; high-resolution, fine-grained visual quality is essential.
>
> To address this, we have **fine-tuned our model at $1024\times 1024$** resolution using **high-resolution ImageNet samples**. As shown in Table A.3, the finetuned model achieves **comparable performance in terms of aesthetics**. We will include updated qualitative comparisons with recent open-source models such as _KOALA_, _PixArt-$\Sigma$_, and _SANA_-series in the final version of the paper to better highlight relative strengths and trade-offs (see Tables B.1 and B.2 below).
>
> We would also like to emphasize that our primary goal is not to replace large-scale commercial models in creative generation tasks, but to demonstrate that **smaller, more reproducible** datasets like **_ImageNet_** can still produce competitive models. There are domains where **prompt fidelity** and **compositional alignment** are the hard constraint and aesthetic quality is irrelevant, examples include medical imaging, rare event generation, or simulation (e.g., autonomous driving). In these contexts, our method matches or outperforms much larger models, despite using **$10\times$ fewer parameters** and more than **$1000\times$ less data**.
>
> # Addressing questions:
>
> ### What resolution is used for SDXL?
>
> Thank you for pointing this out.
>
> The quantitative results reported in Tables 5 (GenEval Benchmark) and 6 (DPG Benchmark) are taken directly from the original papers (e.g., _SDXL_, _SD1.5_, _PixArt_). However, we acknowledge that in the qualitative comparison in Figure 5 _SDXL_ images are generated at $512\times 512$ resolution.
>
> Using a $1024\times 1024$ model fine-tuned using only the **high-resolution subset of ImageNet**, shown in Table A.3, we obtain comparable visual quality compared to its $512\times 512$ counterpart.
>
> In the camera ready version, we will include updated qualitative comparisons at $1024\times 1024$ resolution between our model and _SDXL_.
>
> ### Could the Authors provide more up-to-date baselines
>
> Thank you for the helpful suggestion.
>
> We have updated Tables 5 (GenEval) and 6 (DPGBench) to include comparisons with more recent models that leverage enriched or multi-level captions, including _PixArt-$\Sigma$_, _Janus_, _FLUX_, _SANA_, and _SD3_. (See Table B.1 and B.2 below).
>
> At $512\times 512$ resolution, our models outperform _PixArt-$\Sigma$_ on GenEval, particularly in the **Two Objects**, **Position**, and **Color Attribution** sub-tasks. However, we acknowledge that our models still fall short of the strongest commercial baselines such as _SANA_ and _SD3_.
>
> On DPGBench, our best model again achieves comparable performances to _PixArt-$\Sigma$_ at $512\times 512$ resolution and _Janus_, but lag behind _SANA_, _SD3_ and _FLUX_, which use larger-scale training and commercial infrastructure.
>
> #### Table B.1: Comparison with SOTA models on GenEval Benchmark
> | Model| Nb of params | Training set size | Overall | One obj. | Two obj. | Count. | Col. | Pos. | Col. attr. |
> | --- | --- |:--- | --- | --- | --- | --- | --- | --- | --- |
> | PixArt-$\Sigma$ ($512\times 512$) | 0.6B | 0.035B  | 0.52 | 0.98| 0.59| 0.50| 0.80 | 0.10 | 0.15 |
> | SD v2.1  | 0.9B| 5B+| 0.50  | 0.98  | 0.51| 0.44 | 0.85 | 0.07| 0.17|
> | SDXL| 3.5B| 5B+| 0.55 | 0.98  | 0.74 | 0.39   | 0.85 | 0.15 | 0.23 |
> | SD3 M (512) | 2B| 1B+| 0.62| 0.98 | 0.74 | 0.63   | 0.67 | 0.34 | 0.36|
> | SANA-0.6 | 0.6B | 1B+ | 0.64| 0.99  | 0.71 | 0.63   | 0.91 | 0.16 | 0.42|
> | FLUX-dev | 12B| - | 0.67 | 0.99 | 0.81 | 0.79   | 0.74 | 0.20 | 0.47|
> | DiT-I $512\times 512$ CutMix (Ours) | 0.4B | 0.001B| 0.59 | 0.97 | 0.67     | 0.44   | 0.78 | 0.33 | 0.39 |
>
> #### Table B.2: Comparison with SOTA models on DPG Benchmark
> | Model| Params | Training set size | Global | Entity | Attribute | Relation | Other | Overall |
> | --- | --- | --- | --- | --- | --- | --- | --- | --- |
> | Pixart-$\alpha$ | 0.6B   | 25M | 74.97  | 79.32  | 78.60     | 82.57    | 76.96 | 71.11   |
> | Pixart-$\Sigma$ ($512\times 512$)   | 0.6B   | 35M | 87.5   | 87.1   | 86.5      | 84.0     | 86.1  | 79.5    |
> | Pixart-$\Sigma$ ($1024\times 1024$) | 0.6B   | 35M | 86.9   | 82.9   | 88.9      | 86.6     | 87.7  | 80.5    |
> | Sana-0.6B | 0.6B   | 1B+ | 82.6   | 90.0   | 88.6  | 90.1 | 91.9  | 84.3    |
> | SDXL | 3.5B   | 5B+ | 83.27  | 82.43  | 80.91 | 86.76 | 80.41 | 74.65 |
> | SD3-Medium | 2B | 1B+ | 87.90  | 91.01  | 88.83 | 80.70 | 88.68 | 84.08 |
> | Janus | 1.3B  | 1B+ | 82.33  | 87.38  | 87.70 | 85.46 | 86.41 | 79.68 |
> | FLUX-dev | 12B | -   | 82.1 | 89.5   | 88.7 | 91.1 | 89.4  | 84.0  |
> | DiT-I $256\times 256$ CutMix (Ours) | 0.4B| 1.2M| 82.07  | 85.61 | 84.59     | 91.41    | 74.8  | 77.5 |
> | CAD-I $256\times 256$ CutMix (Ours) | 0.3B| 1.2M| 80.85  | 87.48 | 85.32     | 93.54    | 78.00 | 79.94 |
> | DiT-I $512\times 512$ (Ours) | 0.4B| 1.2M| 79.94  | 83.21  | 83.42 | 90.14    | 72.0  | 75.14 |

---

> > ### Comment · Reviewer_tjpV · 2025-08-04
> > **Official Comment by Reviewer**
> >
> > Thank you for your thoughtful rebuttal and the additional experiments you’ve provided. I appreciate your effort in clarifying several points. That said, a few concerns still remain that I hope you can further address.
> >
> > ### Comparison with LAION-POP Dataset:
> > In Tables A.1 and A.2, the reported aesthetic scores—particularly HPSv2.1 and ImageReward—are notably low. These values make me wonder whether the models have been sufficiently trained to capture aesthetic aspects. Given the current scores, it is a bit difficult to imagine these models being used in practice. Some additional clarification on training quality or underlying factors contributing to these results would be helpful.
> >
> > ### Comparison with State-of-the-Art Models:
> > I understand and respect your position that the goal is not to replace large-scale commercial models, but rather to demonstrate that smaller and more reproducible datasets like ImageNet can still yield competitive results. However, since the rebuttal emphasizes that the model achieves “comparable” performance (e.g., Table A.3), it would further strengthen your claim if aesthetic scores (such as HPSv2.1, ImageReward, PickScore) were directly compared against SoTA models, as done in Tables B.1 and B.2. This would make your argument more convincing and transparent to readers.
> >
> > ### Additional Question – Image Resolution and Research Impact:
> > Have you looked into the average resolution of the original ImageNet images? Many of them appear to be below 512×512, which might limit their usefulness in text-to-image (T2I) tasks that typically benefit from higher-resolution training data. While the reproducibility aspect of using ImageNet is commendable, I would be interested in hearing more about how this work might inspire or enable future T2I research. Could you perhaps elaborate on any concrete downstream use cases or research directions that could benefit from this approach, even if the generated images are not aesthetically strong? Since aesthetics is such a crucial factor in T2I, a clearer discussion on this point would be very valuable.
> >
> > ### Final Remarks – Appreciating the Direction and Openness:
> > I sincerely appreciate the unique direction of your work—exploring how far one can go by training T2I models from scratch using ImageNet. It is a refreshing perspective, especially in an era dominated by large proprietary datasets. At the same time, given ImageNet’s age and relatively low resolution, I remain somewhat uncertain about its practical effectiveness for modern T2I tasks that prioritize visual aesthetics. While your work takes an important step in this direction, some of the concerns around image quality and competitiveness with other open datasets (e.g., LAION-5B or LAION-Aesthetics) still feel only partially addressed.

---

> > > ### Author Response · Authors · 2025-08-05
> > >
> > > We thank the reviewer for their very insightful discussion. As requested, here is the follow up.
> > >
> > > ### Comparison with LAION-POP Dataset on Table A2:
> > >
> > > We would like to clarify that the models in Table A.2 are trained and evaluated at **256² resolution**, which significantly affects aesthetic scores. These metrics were designed for **evaluating high-resolution outputs (e.g., 1024²)**, and their scores degrade considerably at lower resolutions.
> > >
> > > To illustrate this, we extend Table A.2 into Table A.2b (see below) with our DiT-I (Crop) model and KOALA Lightning 700M. We observe a **consistent drop** across both models when evaluated **at lower resolution**. It confirms that the low scores may not fully reflect image quality as they would do for larger images. **We thus consider these scores only for relative comparison between model variants and not in absolute value**.
> > >
> > > #### Table A.2b: Comparison on Aesthetic metrics between our `DiT-I` model and the Koala model. Scores are calculated using `PartiPrompts`.
> > > |Model|Resolution|PickScore|Aes. Score|HPSv2.1|ImageReward|
> > > |---|---|---|:---:|---|---|
> > > |DiT-I Crop|256²|20.63|5.17|0.24|0.12|
> > > |DiT-I Crop|512²|20.94|5.46|0.24|0.20|
> > > |KOALA|256²|17.74|2.90|0.14|-2.22|
> > > |KOALA|512²|20.41|5.07|0.23|-0.42|
> > >
> > > ### Comparison with State-of-the-Art Models:
> > >
> > > Indeed, we agree that we did not provide aesthetics comparison with the state of the art as this was not the focus of our paper. That being said, as we have fine-tuned a model at 1024² resolution on _LAION_POP_ during the rebuttal, we can compare its image quality to SDXL.
> > >
> > > We use the following inference scaling strategy:
> > > - Following SDXL, we use a negative prompt instead of the empy string (`"worst quality, normal quality, low quality, low res, blurry, distortion,[...]"`).
> > > - We sample images among 20 random seeds and 6 CFG values and select the best resulting image according to their HPSv2.1 score.
> > >
> > > The results are in Table A.4 below. Our fine-tuned model achieves **21.6 on PickScore, 6.28 on Aesthetic score, and 0.64 on ImageReward**. This is below but not far behind SDXL, which shows that we indeed exchanged image quality for better compositionality and prompt following. Of course, using HPSv2.1 to select images introduces a bias, and as such we only consider the resulting scores as indicative of the model capabilities.
> > >
> > > In addition, we would like to stress out that our models are **very small compared to the state-of-the-art (400M parameters)** and that generating 1024² images was not the initial goal. We now believe that these models are reaching their limited capacity.
> > >
> > > #### Table A.4: Comparison on Aesthetic metrics between our `DiT-I` model and SDXL at 1024² resolution. Scores are calculated using `PartiPrompts`.
> > > |Model|Inference scaling|PickScore|Aes. Score|HPSv2.1|ImageReward|
> > > |---|---|:---|:---:|---|---|
> > > |DiT-I (LAION-POP-ft)|-|21.04|5.67|0.25|0.24|
> > > |DiT-I (LAION-POP-ft)|$\checkmark$|21.57|6.28|0.29|0.64|
> > > |SDXL|-|22.38|6.55|0.28|0.78|
> > >
> > > ### Additional Question – Image Resolution and Research Impact
> > >
> > > We acknowlegde that ImageNet contains relatively few high-resolution images. Specifically, only 155k images are larger than 512 pixels, and just 26k images larger than 1024. We agree this is a **limitation for training from scratch at high resolution**.
> > >
> > > One possible direction would be to **combine ImageNet with an upscaling network** to generate high-resolution versions. Image augmentations that improve aesthetics may even become a new research avenue.
> > >
> > > Regarding enabling future T2I research, our main goal is to establish a **reproducible and lightweight experimental setup for T2I generation**, akin to what ImageNet represents for class-conditional diffusion research. Recent works such as REPA [1], ReDi [2], and DiS [3] benchmark on class-conditional ImageNet at 256² resolution, and have produced meaningful and widely adopted insights and techniques. We hope our work can enable a similar experimental protocol where **new techniques can be studied without the confounding factor of large-scale noisy data, while being closer to the T2I downstream tasks**.
> > >
> > > [1] Yu et al. Representation Alignment for Generation: Training Diffusion Transformers Is Easier Than You Think. ICLR 2025
> > > [2] Kouzelis et al. Boosting Generative Image Modeling via Joint Image-Feature Synthesis. Arxiv 2025
> > > [3] Fei et al. calable Diffusion Models with State Space Backbone. Arxiv 2025
> > >
> > > ### Final Remarks – Appreciating the Direction and Openness
> > >
> > > We agree that aesthetics is a crucial component of some T2I applications. At the same time, we believe there is growing value in exploring alternative training paradigms that move away from full-scale pretraining on massive, proprietary datasets. Using smaller and fully auditable datasets like ImageNet has the advantages of cost, transparency and reproducibility. Fine-tuning can be done on dedicated dataset such as LAION-Aesthetics to enhance specific capabilities like aesthetic quality.

---

> ### Comment · Reviewer_tjpV · 2025-08-05
> **Response by Reviewer**
>
> Thank you for the clarifications provided in the rebuttal. I appreciate the authors' effort to address the raised concerns. That said, I would like to elaborate on a few remaining points regarding the evaluation and interpretation of your results.
>
> ### Concerns on the KOALA Comparison in Table A.2b:
> The KOALA model was trained at a resolution of 1024×1024 in Table A.4, When it is evaluated at significantly lower resolutions (e.g., 256×256 or 512×512), severe artifacts and distortions may occur. As a result, comparisons made at such low resolutions could unfairly disadvantage KOALA. Since your proposed method is trained only at 256² or 512² resolutions, it naturally performs better under these settings. Therefore, the comparison in Table A.2b does not appear to be a fair evaluation of model quality across architectures.
>
> ### Limited Upper Bound of Performance for ImageNet-trained Models:
> Although the proposed model was fine-tuned(w/o inference scaling) at 1024×1024 resolution, its aesthetic scores still show a noticeable performance gap compared to SDXL in high-resolution evaluations. This suggests that models trained from scratch on ImageNet may face fundamental limitations in terms of scalability and upper-bound quality. As such, I remain uncertain about the broader applicability of the proposed approach in scenarios that demand high visual fidelity.
>
> ### Fair Compositionality Evaluation at High Resolution Against LAION-POP:
> In the rebuttal, you mentioned that “ImageNet-trained models outperformed those trained on LAION-POP in both aesthetic metrics and GenEval” (Table A.1). However, to more convincingly argue for ImageNet’s utility, it would be important to show competitive performance against LAION-POP under the high-resolution fine-tuning setting (e.g., 1024²) in compositionality benchmarks such as Table A.2. Without this, it is difficult to attribute the observed improvements solely to the ImageNet dataset.
> Moreover, since ImageNet lacks native text captions, your strategy to augment captions using an external captioner is not novel—it has already been explored in prior works such as Pixart-Sigma and KOALA. Therefore, **it’s hard to conclude that ImageNet alone contributes meaningfully to improving compositionality.** Given that captions can be synthetically augmented for any dataset, I believe that the quality of the images themselves becomes the most critical factor in T2I research. From this standpoint, the limitations of ImageNet remain quite evident.

---

> > ### Author Response · Authors · 2025-08-07
> >
> > We appreciate the reviewer's insightful comments. As requested, please find our follow-up below.
> >
> > ### Concerns on the KOALA Comparison in Table A.2b
> >
> > As explained in our previous answer, Table A.2b is not meant to compare KOALA against our models. **Rather, it highlights that aesthetic metrics are trained and calibrated for high-resolution images (≥1024²)**, making their results less reliable for lower resolutions like 256². We included KOALA (finetuned at 1024² on an aesthetic-focused dataset) at 256² and 512² only to show these metrics' limitations at low resolution and not to make any claims about KOALA's relative performance. Even top models trained at 1024² score lower when downsampled due to metric calibration.
> >
> > Thus, **Table A.2b supports comparisons only between same model variants for different resolutions, cautioning against cross-resolution metric comparisons**.
> >
> >
> > ### Upper Bound of Performance for ImageNet-trained Models:
> >
> > We respectfully disagree with the importance of artistic performances. High aesthetic quality is not universally important across all T2I applications. **In fields like autonomous driving, robotic simulation or medical image synthesis, realism, prompt fidelity and compositionalily matter more than visual beauty or stylistic appeal**. Models that are designed for artistic content creation are unable to generate realistic pictures of dangerous scenarios for autonomous driving. This does not make the research that led to these models flawed or irrelevant.
> >
> > We recall the focus of our paper and our findings: **contrarily to popular belief, a billion-scale dataset is not required to train high quality text-to-image model**. We show that **ImageNet with careful augmentation** can lead to models that produce good quality pictures (see Figure 1 - maybe not artistic pictures, but good quality nonetheless) and that beat SDXL on compositional benchmarks by about 1% (by an even larger margin, +3% on GenEval and +5% on DPGBench, if we consider 256 pixel resolution). This is not only surprising, it is also useful to the community: current research on T2I is done on proprietary datasets with massive compute that are by definition impossible to reproduce. Our work is exploratory and meant to **lower the barrier of entry for text-to-image research**, not to replace high-end commercial models like SDXL.
> >
> > We answered the question about artistic qualities by fine-tuning a model at 1024² resolution that achieves much higher visual quality than the pictures in Figure 1 (HPSv2.1 0.29 compared to 0.24 originally, ImageReward of 0.64 compared to 0.20 originally). But going further in the direction of larger models, longer training, or aesthetic finetuning is orthogonal to our work. We aim at opening a new direction: establishing a lightweight, reproducible testbed for studying T2I methods from scratch. The community currently lacks such a setup.
> >
> >
> > ### Fair Compositionality Evaluation at High Resolution Against LAION-POP
> >
> > We want to kindly remind the reviewer that no paper has trained a model from scratch on LAION-POP. The KOALA paper did not train their very efficient architecture from scratch on it, as they start from an SDXL checkpoint and then use a very clever distillation loss with SDXL as the teacher.
> >
> > To the best of our knowledge we are the first to train from scratch on LAION-POP (with similar architecture and training budgets) and our result in Table A.1 shows that it is not competitive against ImageNet with our augmentations at 256 resolution. We believe it is unlikely that this performance gap would be closed by finetuning at higher resolution.
> >
> > Besides, there may be a misunderstanding: **we never claimed that ImageNet alone improves compositionality**. We instead shared the surprising discovery that contrarily to popular belief, **a big dataset is not required to get good compositionality**, and that **ImageNet with image (Crop, CutMix) and text augmentations is able to exceed SDXL levels of compositionality**. We think this is valuable research that has good implications for the future as methodological improvements can now be tested **without the cost and the opacity of massive propriatery datasets**.

---

> ### Comment · Reviewer_tjpV · 2025-08-09
> **Official Comment by Reviewer**
>
> From my perspective, the statement that “models designed for artistic content creation are unable to generate realistic pictures of dangerous scenarios for autonomous driving” is **not well-supported**. The LAION-POP and LAION-Aesthetic datasets are not limited to images optimized purely for visual beauty—they also contain a substantial number of highly photo-realistic images, arguably far more than ImageNet. Moreover, the aesthetic scores I emphasize (e.g., HPSv2, ImageReward) are not mere measures of visual beauty; rather, they reflect **human preferences for overall visual quality**, which includes both aesthetic appeal and **realism**. Therefore, it is not accurate to claim that models trained on LAION-based datasets are “unable to generate realistic pictures,” especially when compared to models trained from scratch solely on ImageNet. In fact, as evidenced in Figure 1, the realism of the results does not appear clearly superior.
>
> Second, as I have repeatedly noted, the improvement in compositionality observed with ImageNet is not a result of the raw ImageNet dataset itself, but rather of text augmentation through an image captioner. It is this augmented text that drives the improvement in compositionality metrics. This directly challenges the claim that the proposed method outperforms SDXL in terms of compositionality. SDXL is an older model from 2023 that was not trained with enriched text captions, whereas the proposed approach benefits from augmented text generated by an image captioner. In this context, the observed compositionality improvement cannot be attributed to the underlying ImageNet dataset itself, but rather to the additional caption augmentation. Furthermore, as shown in Table B.2, the compositionality metrics of the proposed method are lower than those of PixArt-Sigma, a more recent text-augmented model. This suggests that the claimed advantage over SDXL does not necessarily extend to stronger, more comparable baselines.
>
> Considering these points, I am not fully convinced that ImageNet with augmented text captions will have a substantial impact on future text-to-image research, particularly when high-quality, large-scale datasets such as LAION already offer higher resolution, greater sample diversity, and enriched captions. While the proposed approach demonstrates interesting results, its broader impact may be more limited than suggested.

---

> > ### Author Response · Authors · 2025-08-09
> >
> > Thank you for the answers. We are keen on having these discussions and we think that overall they help advance research on this field.
> >
> >  ### RE realistic images using models (SDXL) trained on LAION:
> > We experimented with several prompts for several domains:
> > — autonomous driving example “A realistic image of a zebra crossing a city street at night as taken from an autonomous driving vehicle” on the HuggingFace SDXL TPU Demo;
> > — see also the examples and caption in Figure 5 of our submission (eg: An old man with a long grey beard and green eyes)
> >
> > Images generated with SDXL for various downstream  applications are not by default photorealistic but instead artistic.
> >
> > ### RE metrics:
> > To our knowledge, HPSv2 and ImageReward focus on user preferences in terms of aesthetic appeal, visual quality, prompt alignment (appendix B, page 13 of the HPSv2 paper).
> >
> > We are not aware of any reference / work that states that these two metrics measure realism.
> >
> > In fact, for this very reason, the community has recently turned to the “SciScore” [A], a scientific quality and realism metric designed specifically for text-to-image generation (see Figure 1 in [A]).
> >
> > Therefore, we disagree with the reviewer: we consider HPSv2 and ImageReward suitable (and valuable) for measuring “aesthetic appeal” but not “realism”.
> >
> > [A] Science-T2I: Addressing Scientific Illusions in Image Synthesis, CVPR 2025
> >
> > ### RE claims on reaching state-of-the-art performances and compositionality:
> > Our claim is “one can match or outperform models trained on massive web-scraped collections, using only ImageNet enhanced with well-designed text and image augmentations”
> >
> > SDXL is trained on more than 1B+ images.
> > Instead, we match these performances “while using just 1/10th the parameters and 1/1000th the training images”.
> >
> > ### RE impact:
> > Impact is not easily quantifiable.
> >
> > We believe in and advocate for open, reproducible science. We consider that our work has impact if at least one lab (eg academic where typically resources are limited) is considers training their own model on open, smaller-scale data with simple augmentations instead of relying on massive decaying datasets and billion-parameter models.
> >
> > We think that our work opens the way to do so.
> >
> > We understand that our findings go against the current hype: “bigger is better”. This is indeed surprising and we believe that it is worth spreading.

---

### Note · Authors · 2025-08-12

We thank the reviewers and AC for their constructive engagement, which helped clarify and strengthen our work.

In an era **dominated by large-scale dataset training**, we show that it’s possible to train an open, reproducible, and safe (free of inappropriate content) text-to-image model, achieving quality comparable to SOTA, using **1000× less data** (ImageNet + text and image augmentations) and **10× fewer parameters**.

Reviewers praised our method's clarity, simplicity, and timeliness: **fyFz**: “*a step toward open science*”; **qDs6**: “*lowering the entry barrier*”; **tjpV**: “*a refreshing perspective*”.


In the rebuttal we addressed:

**ImageNet vs other datasets (fyFz, tjpV)**

We experimented with 1M random DataComp-1B images (**fyFz**; Tab D.3) and LAION-POP (**tjpV**; Tab A.2), but ImageNet performed better due to its **balanced, diverse distribution**. Its **non-decaying** nature ensures our results are fully reproducible.

**Fine-tuning on other datasets (tjpV, fyFz)**

We demonstrated that pretraining on ImageNet followed by fine-tuning on targeted datasets like LAION-POP is a Pareto-optimal strategy, leading to **improved aesthetic scores** (Tab A.3). **fyFz** acknowledged that their question was addressed.

**Improving on non-object centric and human specific prompts (fyFz, he9B)**

We conducted an experiment training from scratch on ImageNet+MSCOCO, which introduced **non-object-centric and human-specific generation capabilities** without affecting performance (Tab D.3). Both reviewers noted that their concerns had been addressed.

**Robustness (qDs6)**

Experiments with the Flow Matching paradigm (Tab B.3) and CAD-I crop augmentation (Tab B.4) confirm our method’s **robustness**. **qDs6** agreed these additions directly addressed their concerns.

**Divergence and Alignment**

While **tjpV** remained partial to higher aesthetic quality and skeptical of ImageNet, we clarified our aim is **accessibility and reproducibility**. Experimental concerns were addressed, and other reviewers endorsed our approach.

These updates address concerns about comparisons with LAION-POP and random samples from larger datasets, robustness across architectures and paradigms, and mitigating ImageNet’s object-centric, non-human bias. Reviewers acknowledged the improvements, noting all questions (fyFz) and concerns (he9B, qDs6) were resolved.

We will incorporate these points, typo fixes, and citations in the final version to support our open-science approach.

---

### Decision · Program_Chairs · 2025-09-17

**Decision:**

Reject

**Comment:**

Four reviewers went over this submission. The reviewers raised concerns about:
- the novelty of the contribution [tjpV, qDs6, he9B]
- the novelty of the findings (potential of small scale dataset to achieve state-of-the-art)  [tjpV]
- the claims in the paper (not well supported by experimental evidence) [tjpV, fyFz]
- the unconvincing validation and unclear impact of the presented results [tjpV, qDs6, he9B, fyFz] (comparisons wrt models trained on high quality datasets, no human evaluations, fairness of comparisons, no comparisons with recent models, low resolution experiments, complex scenes)
- the practical quality of generated images [tjpV]
- the quality of the presentation [qDs6]

The rebuttal partially addressed the reviewers concerns. In particular, the rebuttal introduced:
- new experimental evidence contrasting with work training/finetuning text-to-image models on small scale data
- comparisons with more recent text-to-image models
- results on complex scenes (extending ImageNet with COCO or LAION-POP).

The rebuttal also introduced additional metrics (as opposed to user studies) to capture human preference (ImageReward and HPSv2) and argued about the novelty ("demonstrating that a reproducible, resource-efficient training on a small, open dataset like ImageNet can achieve state-of-the-art performance").

During discussion, many reviewers engaged with the authors. After rebuttal, the recommendation remains borderline. The AC engaged in discussion with the reviewers. The reviewers agreed (despite their final ratings) on the following discussion points: 1) importance of aesthetic quality, 2) the upper bound limitation brought by ImageNet, 3) the claims of the paper which still appear too strong given the experimental evidence. There were disagreements among reviewers on the novelty and clarity of presentation.

The authors reached out to the AC about fundamental disagreements with some parts of the feedback received, and summarized the key points of the process in the final remarks. The AC took these messages into consideration when discussing and making the final recommendation.

The AC agrees with the reviewers that the claims are too strong given the evidence, since the claimed advantages/competitiveness do not extend to stronger baselines. The paper would benefit from rethinking its current narrative and from adjusting the claims to the reported results. Given the current results, the significance of the contribution remains unclear. Therefore, the AC recommends to reject.